# Rectified Point Flow:
# Generic Point Cloud Pose Estimation

**Tao Sun**[*]
Stanford University

**Liyuan Zhu**[*]
Stanford University

**Shengyu Huang**
NVIDIA Research

**Shuran Song**
Stanford University

**Iro Armeni**
Stanford University

## Abstract

We present *Rectified Point Flow*, a unified parameterization that formulates pairwise point cloud registration and multi-part shape assembly as a single conditional generative problem. Given unposed point clouds, our method learns a continuous point-wise velocity field that transports noisy points toward their target positions, from which part poses are recovered. In contrast to prior work that regresses partwise poses with ad-hoc symmetry handling, our method intrinsically learns assembly symmetries without symmetry labels. Together with an overlap-aware encoder focused on inter-part contacts, Rectified Point Flow achieves a new state-of-the-art performance on six benchmarks spanning pairwise registration and shape assembly. Notably, our unified formulation enables effective joint training on diverse datasets, facilitating the learning of shared geometric priors and consequently boosting accuracy. Our code and models are available at *https://rectified-pointflow.github.io/*.

## 1   Introduction

Estimating the relative poses of rigid parts from 3D point clouds for alignment is a core task in computer vision and robotics, with applications spanning pairwise registration [1] and complex multi-part shape assembly [2]. In many settings, the input consists of an unordered set of part-level point clouds–without known correspondences, categories, or semantic labels–and the goal is to infer a globally consistent configuration of poses, essentially solving a multi-part (two or more) point cloud pose estimation problem. While conceptually simple, this problem is technically challenging due to the combinatorial space of valid assemblies and the prevalence of symmetry and part interchangeability in real-world shapes [3, 4, 5].

Despite sharing the goal of recovering 6-DoF transformations, different 3D reasoning tasks—such as object pose estimation, part registration, and shape assembly—have historically evolved in silos, treating each part independently and relying on task-specific assumptions and architectures. For instance, object pose estimators often assume known categories or textured markers [6, 7], while part assembly algorithms may require access to a canonical target shape or manual part correspondences [8]. This fragmentation has yielded solutions that perform well in narrow domains but fail to generalize across tasks, object categories, or real-world ambiguities.

Among these tasks, multi-part shape assembly presents especially unique challenges. The problem is inherently under constrained: parts are often symmetric [9], interchangeable [10], or geometrically ambiguous, leading to multiple plausible local configurations. As a result, conventional part-wise registration can produce flipped or misaligned configurations that are locally valid but globally inconsistent with the intended assembly. Overcoming such ambiguities requires a model that can

---

[*]Equal contribution.

39th Conference on Neural Information Processing Systems (NeurIPS 2025).

reason jointly about part identity, relative placement, and overall shape coherence—without relying on strong supervision or hand-engineered heuristics.

In this work, we revisit 3D pose regression and propose a generative approach for generic point cloud pose estimation that casts the problem as learning a continuous point-wise flow field over the input geometry, effectively capturing priors over assembled shapes. Our method, Rectified Point Flow, models the motion of points from random Gaussian noise in Euclidean space toward the point clouds of assembled objects. This learned flow implicitly encodes part-level transformations, enabling both discriminative pose estimation and generative shape assembly within a single framework. Rectified Point Flow consists of an encoder that extracts point-wise features, and a flow model that, given the features, predicts the final assembled positions.

To instill geometric awareness of inter-part relationships, we pretrain the encoder on large-scale 3D shape datasets: predicting point-wise overlap across parts, formulated as a binary classification task. While GARF [11] also highlights the value of encoder pretraining for a flow model, it relies on mesh-based physical simulation [12] to generate fracture-based supervision signals. In contrast, we introduce a lightweight and scalable alternative that constructs pretraining data by computing geometric overlap between parts. Our data generation is agnostic to data sources tailored for different tasks—including part segmentation [13, 14, 15], shape assembly [12, 16, 17], and registration [18, 19]—without requiring watertight mesh or simulation, an important step towards scalable pretraining for pose estimation.

Our flow-based pose estimation departs from traditional pose-vector regression in three ways: (i) **Joint shape-pose reasoning**: We cast the registration and assembly tasks as one unified task that reconstructs the complete shape while simultaneously enabling the estimation of part poses; (ii) **Scalable shape prior learning**: By training to predict the final assembled point cloud, our model learns from heterogeneous datasets and part definitions, yielding scalable training and transferable geometric knowledge across standard pairwise registration, fracture reassembly, and complex furniture assembly tasks; and (iii) **Intrinsic symmetry handling**: Rather than regressing pose vectors directly in $\mathrm{SE}(3)$ space, we operate in Euclidean space over dense point clouds. This makes the model inherently robust to symmetries, part interchangeability, and spatial ambiguities that often challenge conventional methods. Our main contributions are summarized as follows:

- We propose Rectified Point Flow, a generative approach for generic point cloud pose estimation that addresses both pairwise registration and multi-part assembly tasks and achieves state-of-the-art performances on all the tasks.

- We propose a generalizable pretraining strategy with geometric awareness of inter-part relationships across several 3D shape datasets, and formulate it as point-wise overlap prediction.

- We show that our parameterization supports joint training across different registration tasks, boosting the performance on each individual task.

## 2 Related Work

**Parametrization for Pose Estimation.** Euler angles and quaternions are the predominant parametrization of rotation in various pose regression tasks [20, 21, 22, 23, 24, 25, 11, 26] due to their simplicity and usability. As Euler angles and quaternions are discontinuous representations, Zhou *et al.* [27] proposed to represent 3D rotation with a continuous representation for neural networks using 6D and 7D vectors. In contrast to directly regressing pose vectors, other methods train networks to find sparse correspondences between image pairs or point cloud pairs and extract pose vectors using Singular Value Decomposition (SVD) [28, 29, 8, 30, 31, 32]. More recently, RayDiffusion [33] proposed to represent camera poses as ray bundles, naturally suited for coupling image features and transformer architectures. Huang *et al.* [34] adopted a point cloud generative model for policy learning in robot pick-and-place tasks, then recovered relative poses between the object and gripper via SVD. DUSt3R [35] directly regresses the pointmap of each camera in a global reference frame and then extracts the camera pose using RANSAC-PnP [36, 37]. Our proposed Rectified Point Flow, extends the dense point cloud or point map representation for learning generalizable pose estimation on point cloud registration and shape assembly tasks.

**Learning-based 3D Registration.** 3D registration aims to align point cloud pairs in the same reference frame by solving the relative transformation from source to target. The first line of work

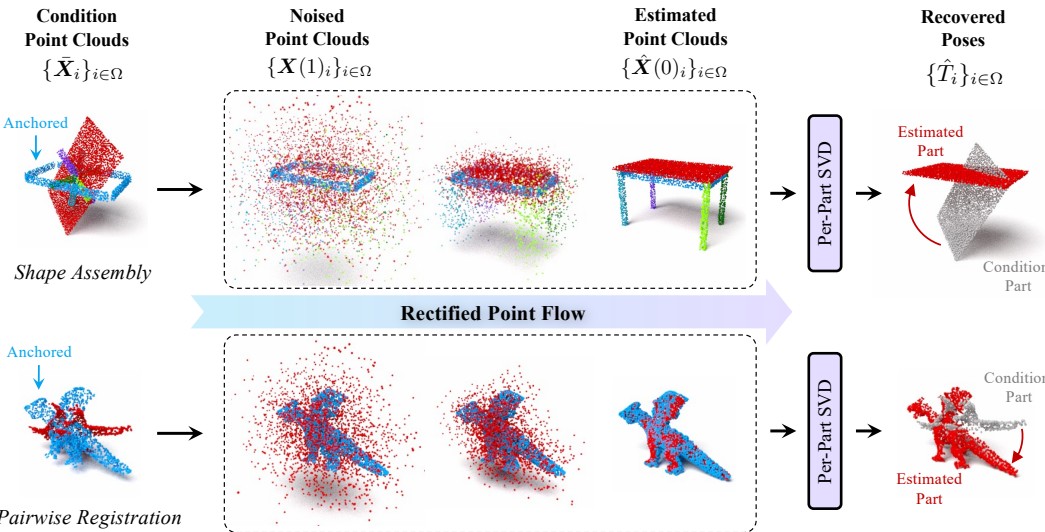

Figure 1: **Rectified Point Flow's Pose-from-Shape Pipeline**. Our formulation supports *shape assembly* (first row) and *pairwise registration* (second row) tasks in a single framework. Given a set of unposed part point clouds $\{\bar{X}_i\}_{i\in\Omega}$, Rectified Point Flow predicts each part's point cloud at the target assembled state $\{\hat{X}_i(0)\}_{i\in\Omega}$. Subsequently, we solve Procrustes problem via SVD between the condition point cloud $\bar{X}_i$ and the estimated point cloud $\hat{X}_i(0)$ to recover the rigid transformation $\hat{T}_i$ for each non-anchored part.

focuses on correspondence-based methods [38, 1, 39, 40] that first extract correspondences between point clouds, followed by robust estimators to recover the transformation. Subsequent works [29, 41, 42, 32] advance the performance by learning more powerful features with improved architecture and loss design. The second line of work comprises direct registration methods [31, 43, 30, 22] that directly compute a score matrix and apply differentiable weighted SVD to solve for the transformation. Correspondence-based methods can fail in extremely low-overlap scenarios in shape assembly and direct methods fall short in terms of pose accuracy. Our method, which directly regresses the coordinates of each point in the source point cloud, is agnostic and more generalizable to varying overlap ratios compared to direct methods.

**Multi-Part Registration and Assembly.** Multi-part registration and shape assembly generalize pairwise relative pose estimation to multiple parts, with applications in furniture assembly [17] and shape reassembly [12]. Methods [44, 45, 26, 46, 47] tackle the multi-part registration problem by estimating the transformation for each rigid part in the scene (multi-source and multi-target). Multi-part shape assembly differs as a task from registration because it has multi-source input and a canonical target, and each part has almost 'zero' overlap w.r.t. each other. Chen *et al.* [48] adopt an adversarial learning scheme to examine the plausibility for different shape configurations. Wu *et al.* [49] leverage SE(3) equivariant representation to handle pose variations in shape assembly. DiffAssembly [50] and PuzzleFusion [24, 25] leverage diffusion models to predict the transformation for each part. GARF [11] combines fracture-aware pretraining with a flow matching model to predict per-part transformation. These methods, however, do not handle interchangeability and symmetry as well as ours does. Moreover, Rectified Point Flow is the first solution for furniture assembly of 3D shapes on the PartNet-Assembly [15] and IKEA-Manual [17] datasets.

## 3 Pose Estimation via Rectified Point Flow

Rectified Point Flow addresses the multi-part point cloud pose estimation problem, defined in Sec. 3.1. The overall pipeline consists of two consecutive stages: overlap-aware point encoding (Sec. 3.2) and conditional Rectified Point Flow (Sec. 3.3). Finally, we explain how our formulation inherently addresses the challenges posed by symmetric and interchangeable parts in Sec. E.

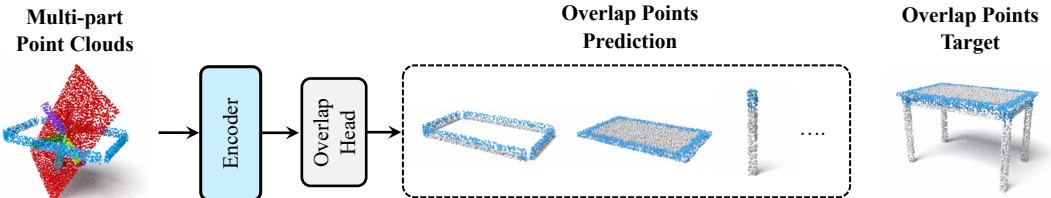

**Multi-part Point Clouds**  **Overlap Points Prediction**  **Overlap Points Target**

Figure 2: **Encoder pre-training via overlap points prediction.** Given unposed multi-part point clouds, our encoder with a point-wise overlap prediction head performs a binary classification to identify overlapping points. Predicted overlap points are shown in blue. For comparison, the ground-truth overlap points are visualized on the assembled object for clarity (target overlap).

## 3.1 Problem Definition

Consider a set of unposed point clouds of multiple object parts, $\{X_i \in \mathbb{R}^{3 \times N_i}\}_{i \in \Omega}$, where $\Omega$ is the part index set, $H := |\Omega|$ is the number of parts, and $N_i$ is the number of points in part $i$. The goal is to solve for a set of rigid transformations $\{T_i \in \mathrm{SE}(3)\}_{i \in \Omega}$ that align each part in the unposed multi-part point cloud $X$ to form a single, assembled object $Y$ in a global coordinate frame, where

$$X := \bigcup_{i \in \Omega} X_i \in \mathbb{R}^{3 \times N}, \quad Y := \bigcup_{i \in \Omega} T_i X_i \in \mathbb{R}^{3 \times N}, \quad \text{and } N := \sum_{i \in \Omega} N_i. \tag{1}$$

To eliminate global translation and rotation ambiguity, we set the first part ($i = 0$) as the anchor and define its coordinate frame as the global frame. All other parts are registered to this anchor.

## 3.2 Overlap-aware Point Encoding

Pose estimation relies on geometric cues from mutually overlapping regions among connected parts [29, 32, 11]. In our work, we address this challenge through a pretraining module that develops a task-agnostic, overlap-aware encoder capable of producing pose-invariant point features. As illustrated in Fig. 2, we train an encoder $F$ to identify overlapping points in different parts. Given a set of unposed parts $\{X_i\}_{i \in \Omega}$, we first apply random rigid transforms $\tilde{T}_i \in \mathrm{SE}(3)$ and compose transformed point clouds $\tilde{X}_i = \tilde{T}_i X_i$ as input to the encoder. These data augmentations enable the encoder to learn more robust pose-invariant features. The encoder then computes per-point features $C_{i,j} \in \mathbb{R}^d$ for the $j$-th point on part $i$, after which an MLP overlap prediction head estimates the overlap probability $\hat{p}_{i,j}$. The binary ground-truth label $p_{i,j}$ is 1 if point $\tilde{x}_{i,j}$ falls within radius $\epsilon$ of at least one point in other parts.

We train both the encoder and the overlap head using binary cross-entropy loss. For objects without predefined part segmentation, we employ off-the-shelf 3D part segmentation methods to generate the necessary labels. The features extracted by our trained encoder subsequently serve as conditioning input for our Rectified Point Flow model.

## 3.3 Generative Modeling for Pose Estimation

The overlap-aware encoder identifies potential overlap regions between parts but cannot determine their final alignment, particularly in symmetric objects that allow multiple valid assembly configurations. To address this limitation, we formulate the point cloud pose estimation as a *conditional generation task*. With this approach, Rectified Point Flow leverages the extracted point features to sample from the conditional distribution of all feasible assembled states across multi-part point clouds, generating estimates that maximize the likelihood of the conditional input point cloud. By recasting pose estimation as a generative problem, we naturally accommodate the inherent ambiguities arising from symmetry and part interchangeability in the data.

**Preliminaries.** Rectified Flow (RF) [51, 52] is a score-free generative modeling framework that learns to transform a sample $X(0)$ from a source distribution, into $X(1)$ from a target distribution. The forward process is defined as linear interpolation between them with a timestep $t$ as

$$X(t) = (1 - t)X(0) + tX(1), \quad t \in [0, 1]. \tag{2}$$

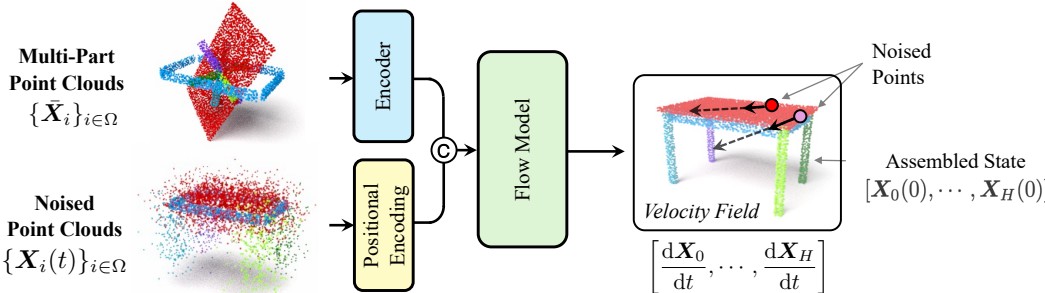

Figure 3: **Learning Rectified Point Flow.** The input to Rectified Point Flow are the condition point clouds $\{\tilde{X}_i\}_{i\in\Omega}$ and noised point clouds $\{X_i(t)\}_{i\in\Omega}$ at timestep $t$. They are first encoded by the pre-trained encoder and the positional encoding, respectively. The encoded features are concatenated and passed through the flow model, which predicts per-point velocity vectors $\{dX_i(t)/dt\}_{i\in\Omega}$ and defines the flow used to predict the part point cloud in its assembled state.

The reverse process is modeled as a velocity field $\nabla_t X(t)$, which is parameterized as a network $V(t, X(t) \mid X)$ conditioned on $X$ and trained using conditional flow matching (CFM) loss [53],

$$\mathcal{L}_{\text{CFM}}(V) = \mathbb{E}_{t,X}\left[\|V(t, X(t) \mid X) - \nabla_t X(t)\|^2\right]. \tag{3}$$

**Rectified Point Flow.** In our method, we directly apply RF to the 3D Euclidean coordinates of the multi-part point clouds. Let $X_i(t) \in \mathbb{R}^{3 \times M_i}$ denote the time-dependent point cloud for part $i$, where $M_i$ is number of sampled points. At $t = 0$, $\{X_i(0)\}_{i\in\Omega}$ is uniformly sampled from the assembled object $Y$, while at $t = 1$, points on each part are independently sampled from a Gaussian, *i.e.*, $X_i(1) \sim \mathcal{N}(0, I)$. Then, we define the continuous flow for each part as straight-line interpolation in 3D Euclidean space between the points in noised and assembled states. Specifically, for each part $i$,

$$X_i(t) = (1 - t)X_i(0) + tX_i(1), \quad t \in [0, 1]. \tag{4}$$

The velocity field of Rectified Point Flow is therefore,

$$\frac{dX_i(t)}{dt} = X_i(1) - X_i(0). \tag{5}$$

We fix the anchored part ($i = 0$) by setting $X_0(t) = X_0(0)$ for all $t \in [0, 1]$, implemented via a mask that zeros out the velocity for its points. Once the model predicts the assembled point cloud of each part $\hat{X}_i(0)$, we recover its pose $T_i$ in a Procrustes problem,

$$\hat{T}_i = \underset{\hat{T}_i \in \text{SE}(3)}{\arg\min} \|\hat{T}_i X_i - \hat{X}_i(0)\|_F. \tag{6}$$

Solving poses $\hat{T}_i$ for all non-anchored parts via SVD completes the pose estimation task in Eq. 1.

**Learning Objective.** We train a flow model $V$ to recover the velocity field in Eq. 5, taking the noised point clouds $\{X_i(t)\}_{i\in\Omega}$ and conditioning on unposed multi-part point cloud $X$, as shown in Fig. 3. First, we encode $X$ using the pre-trained encoder $F$. For each noised point cloud, we apply a positional encoding to its 3D coordinates and part index, concatenate these embeddings with the point features, and feed the result into the flow model. We denote its predicted velocity field by the flow model for all points by $V(t, \{X_i(t)\}_{i\in\Omega}; X) \in \mathbb{R}^{3 \times M}$. We optimize the flow model $V$ by minimizing the conditional flow matching loss in Eq. 3.

### 3.4 Invariance Under Rotational Symmetry and Interchangeability

In our method, the straight-line point flow and point-cloud sampling, while simple, guarantee that every flow realization and its loss in Eq. (3) remain invariant under an assembly symmetry group $\mathcal{G}$:

**Theorem 1** ($\mathcal{G}$-invariance of the learning objective). *For every element $g \in \mathcal{G}$, we have the learning objective in Eq. 3 following $\mathcal{L}_{\text{CFM}}(V) = \mathcal{L}_{\text{CFM}}(g(V(t, \{X_i(t)\}_{i\in\Omega}; g(X))))$.*

The formal definition of $\mathcal{G}$ and the proof of Theorem 1 appear in the supplementary material. As a result, the flow model learns all the symmetries in $\mathcal{G}$ during training, without the need for additional hand-made data augmentation or heuristics on symmetry and interchangeability.

# 4 Experiments

**Implementation Details.** We use PointTransformerV3 (PTv3) [54] as the backbone for point cloud encoder, and use Diffusion Transformer (DiT) [55] as our flow model. Each DiT layer applies two self-attention stages: *(i)* part-wise attention to consolidate part-awareness, and *(ii)* global attention over all part tokens to fuse information. We stabilize the attention computation by applying RMS Normalization [56, 57] to the query and key vectors per head before attention operations. We sample the time steps from a U-shaped distribution following [58]. We pre-train the PTv3 encoder on all datasets with an additional subset of Objaverse [14] meshes, where we apply PartField [13] to obtain annotations. After pretraining, we freeze the weights of the encoder. We train our flow model on 8 NVIDIA A100 80GB GPUs for 400k iterations with an effective batch size of 256. We use the AdamW [59] optimizer with an initial learning rate $5 \times 10^{-4}$ which is halved every 25k iterations after the first 275k iterations.

Table 1: **Dataset statistics.** We train our flow model on six datasets with varying sizes, part definitions, and complexities. The encoder is pre-trained on these datasets with an extra Objaverse dataset.

| Dataset | Task | Part Definition | Train & Val | | Test | |
|---|---|---|---|---|---|---|
| | | | # Samples | # Parts | # Samples | # Parts |
| IKEA-Manual [17] | Assembly | Reusability and packing | 84 | [2, 19] | 18 | [2, 19] |
| TwoByTwo [16] | Assembly | Insertable parts | 308 | [2, 2] | 144 | [2, 2] |
| PartNet-Assembly | Assembly | Semantics and functions | 23755 | [2, 64] | 261 | [2, 64] |
| BreakingBad [12] | Assembly | Fracture simulation | 35114 | [2, 49] | 265 | [2, 49] |
| TUD-L [18] | Registration | RGB-D sensor scans | 19138 | [2, 2] | 300 | [2, 2] |
| ModelNet-40 [19] | Registration | Random partition | 19680 | [2, 2] | 260 | [2, 2] |
| Objaverse 1.0 [14] | *Pre-training* | *From PartField [13]* | 63199 | [3, 12] | 6794 | [3, 12] |

## 4.1 Experimental Setting

**Datasets.** For the multi-part shape assembly task, we experiment on the BreakingBad [12], TwoByTwo [16], PartNet [15], and IKEA-Manual [17] datasets. The PartNet dataset has been processed for the shape assembly task following the same procedure as [17] but includes all object categories; we refer to this version as PartNet-Assembly. Evaluation of the pairwise registration is performed on the TUD-L [18] and ModelNet-40 [19] datasets. We follow [22] for prepossessing the TUD-L dataset. We split all datasets into train/val/test sets following existing literature for fair comparisons. These datasets define parts at distinct levels, ranging from random partitions (e.g., ModelNet-40 and BreakingBad) to human-labeled (e.g., semantically meaningful parts in PartNet and IKEA-Manual). The statistics and information of all datasets are summarized in Tab. 1.

**Evaluation Protocols.** We evaluate the pose accuracy following the convention of each benchmark, with Rotation Error (RE), Translation Error (TE), Rotation Recall at 5° (Recall @ 5°), and Translation Recall at 1 cm (Recall @ 1 cm). For the shape assembly task, we measure Part Accuracy (Part Acc) by computing per object the fraction of parts with Chamfer Distance under 1 cm, and then averaging those per-object scores across the dataset, following [25, 11, 30, 17].

Following [11], we select the largest-volume part as the anchor and fix it during inference. However, this effectively provides the model with anchor pose in the object's CoM (center of mass) frame, an unrealistic assumption for real-world assembly applications. Therefore, we also train our model in an anchor-free setting (see Appendix B: Anchor-free Models), and argue that anchor-free evaluation should be the standard protocol for shape assembly tasks.

**Baseline Methods.** We evaluated our method against state-of-the-art methods for pairwise registration and shape assembly. For pairwise registration, we compare against DCPNet [31], RPMNet [30], GeoTransformer [32], and Diff-RPMNet [22]. For shape assembly, we compare against MSN [48], SE(3)-Assembly [49], Jigsaw [60], PuzzleFussion++ [25], and GARF [11]. We report our performances under two training configurations: dataset-specific training where models are trained independently for each dataset (denoted *Ours (Single)*), and joint training where a single model is trained across all datasets (denoted *Ours (Joint)*).

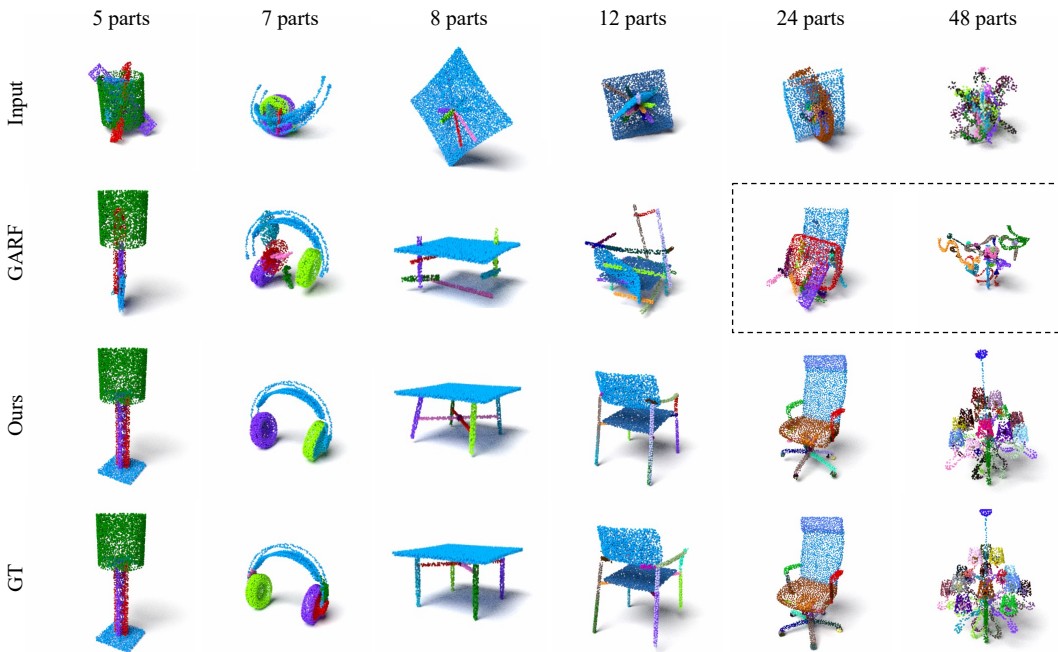

Figure 4: **Qualitative Results on PartNet-Assembly.** Columns show objects with increasing number of parts (left to right). Rows display (1) colored input point clouds of each part, (2) GARF outputs (dashed boxes indicate samples limited to 20 by GARF's design, selecting the top 20 parts by volume), (3) Rectified Point Flow outputs, and (4) ground-truth assemblies. Compared to GARF, our method produces more accurate pose estimation on most parts, especially as the number of parts increases.

Table 2: **Multi-Part Assembly Results.** Rectified Point Flow (Ours) achieves the best performance across all metrics on BreakingBad-Everyday, TwoByTwo, and PartNet-Assembly datasets.

| Methods | BreakingBad-Everyday [12] | | | TwoByTwo [16] | | PartNet-Assembly | | |
| --- | --- | --- | --- | --- | --- | --- | --- | --- |
| | RE ↓ [deg] | TE ↓ [cm] | Part Acc ↑ [%] | RE ↓ [deg] | TE ↓ [cm] | RE ↓ [deg] | TE ↓ [cm] | Part Acc ↑ [%] |
| MSN [48] | 85.6 | 15.7 | 16.0 | 70.3 | 28.4 | – | – | – |
| SE(3)-Assembly [49] | 73.3 | 14.8 | 27.5 | 52.3 | 23.3 | – | – | – |
| Jigsaw [60] | 42.3 | 10.7 | 68.9 | 53.3 | 36.0 | – | – | – |
| PuzzleFusion++ [25] | 38.1 | 8.0 | 76.2 | 58.2 | 34.2 | – | – | – |
| GARF [11] | 9.9 | 2.0 | 93.0 | 22.1 | 7.1 | 66.9 | 21.9 | 25.7 |
| ***Ours** (Single)* | 9.6 | **1.8** | **93.5** | 18.7 | 4.1 | 24.8 | 15.4 | 50.2 |
| ***Ours** (Joint)* | **7.4** | 2.0 | 91.1 | **13.2** | **3.0** | **21.8** | **14.8** | **53.9** |

## 4.2 Evaluation

We report pose accuracy for shape assembly and pairwise registration in Tab. 2 [2] and Tab. 3, respectively. Our model outperforms all existing approaches by a substantial margin. For **multi-part assembly**, the closest competitor is GARF [11], which formulates per-part pose estimation as 6-DoF pose regression; see Figs. 4 and 5. We attribute our superior results to two key advantages of Rectified Point Flow: (i) in contrast to our closest competitor GARF [11] which performs 6-DoF pose regression, our dense shape-and-pose parametrization helps the model learn better global shape prior and fine-grained geometric details more effectively; and (ii) our generative formulation natively handles

---

[2]We found that the BreakingBad benchmark [12, 11, 25] originally computed rotation error (RE) using the RMSE of Euler angles, which is not a proper metric on SO(3). To ensure consistency, we re-evaluate all baselines using the geodesic distance between rotation matrices via the Rodrigues formula [61, 31, 29, 32]. For *Ours (Single)* on TwoByTwo, we used encoder pretrained in the *Ours (Joint)* setting but trained the flow model only on TwoByTwo, due to its limited size.

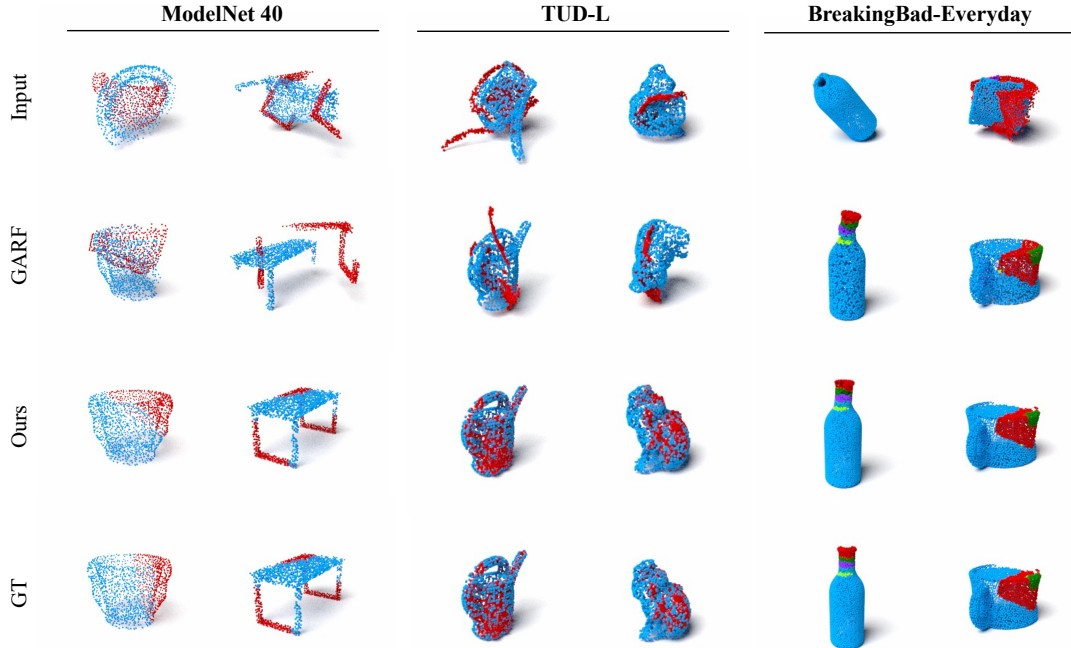

Figure 5: **Qualitative Results Across Registration and Assembly Tasks.** From left to right: pairwise registration on ModelNet 40 and TUD-L, shape assembly on BreakingBad-Everyday. From top to bottom: Colored input point clouds, GARF results, ours, and ground truth (GT). Our single model performs the best across registration and assembly tasks.

Table 3: **Pairwise Registration Results.** Rectified Point Flow (Ours) outperforms all baselines on both TUD-L and ModelNet 40, achieving the highest accuracy and lowest errors across all metrics.

| Methods | TUD-L [18] | | ModelNet 40 [19] | |
|---|---|---|---|---|
| | Recall @5° ↑ [%] | Recall @1cm ↑ [%] | RE ↓ [deg] | TE ↓ [unit] |
| DCPNet [31] | 23.0 | 4.0 | 11.98 | 0.171 |
| RPMNet [30] | 73.0 | 89.0 | 1.71 | 0.018 |
| GeoTransformer [32] | 88.0 | 97.5 | 1.58 | 0.018 |
| GARF [11] | 53.1 | 52.5 | 42.5 | 0.063 |
| Diff-RPMNet [22] | 90.0 | 98.0 | – | – |
| ***Ours** (Single)* | 97.0 | 98.7 | 1.37 | 0.003 |
| ***Ours** (Joint)* | **97.7** | **99.0** | **0.93** | **0.002** |

part symmetries and interchangeability. For **pairwise registration**, GARF–despite being retrained on target datasets–fails to generalize beyond the original task. In contrast, our method achieves a new state-of-the-art performance on registration benchmarks, outperforming methods designed solely for registration (*e.g.*, GeoTransformer [32] and Diff-RPMNet [22]) and demonstrating strong generalization across different datasets (Fig 5). We also achieve the strongest shape assembly performance on IKEA-Manual [17]; for more details on evaluation and visualizations, see supplementary.

**Joint Training.** By recasting pairwise registration as a two-part assembly task, our unified formulation enables joint training of the flow model on all six datasets—including very small sets like TwoByTwo (308 samples) and IKEA-Manual (84 samples)—and the additional pretraining data from Objaverse. *Ours (Joint)* consistently matches or outperforms the dataset-specific (*Ours (Single)*) models. For example, on TwoByTwo the rotation error drops from 18.7° to 13.2° (≈30%), and on BreakingBad from 9.6° to 7.4° (≈23%), while on ModelNet40, the rotation error is reduced from 1.37° to 0.93°. These results demonstrate that joint training enables the model to learn shared geometric priors from datasets with diverse part segmentation, symmetries, and common pose distributions, which substantially boosts performance, particularly on datasets with limited training samples.

**Symmetry Handling.** We demonstrate our model's ability to handle symmetry (Sec. E) on IKEA-Manual [17], a dataset with symmetric assembly configurations. As shown in Fig. 6, while being only trained on a single configuration (left), Rectified Point Flow samples various reasonable assembly configurations (right), conditioned on the same input unposed point clouds. Note how our model permutes the 12 repetitive vertical columns and swaps the 2 middle baskets, yet always retains the non-interchangeable top and footed bottom baskets in their unique positions.

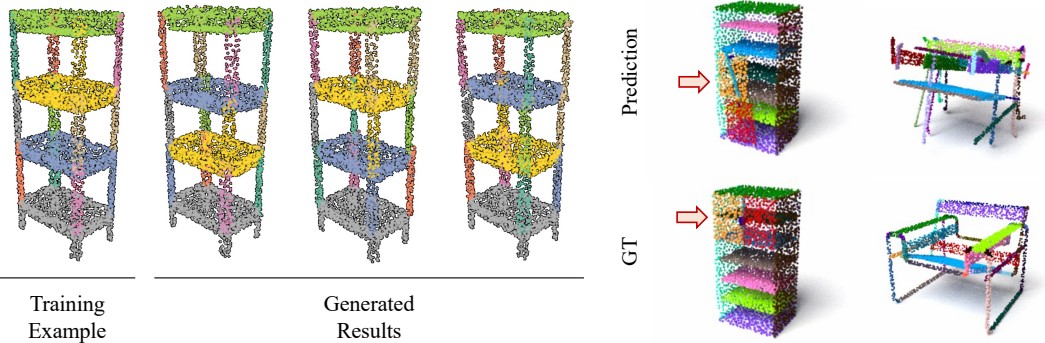

Figure 6: **Learning from a symmetric assembly.** Left to right: (1) a single training sample from IKEA-Manual [17], and (2–4) three independent samples generated, conditioned on the same inputs. Parts are color-coded consistently across plots. (*Best viewed in color.*)

Figure 7: **Two common failure types.** First column: Assemblies that are geometrically plausible but mechanically non-functional. Second column: Objects with high geometric complexity.

Table 4: **Ablation on Encoder Pre-training.** We ablate the impact of different pre-training tasks on the shape assembly performance. Our overlap detection pre-training yields the best results.

| Dataset | Encoder | Pre-training Task | RE ↓ [deg] | TE ↓ [cm] | Part Acc ↑ [%] |
|---|---|---|---|---|---|
| BreakingBad-Everyday [12] | MLP | *No Pre-training* | 41.7 | 12.3 | 68.3 |
| | PTv3 [54] | *No Pre-training* | 18.5 | 4.9 | 79.5 |
| | PTv3 [54] | Instance Segmentation | 16.7 | 4.4 | 80.9 |
| | PTv3 [54] | Overlap Detection (*ours*) | **9.6** | **1.8** | **93.5** |
| PartNet-Assembly [15] | Point-BERT [62] | Point Cloud Completion | 27.4 | 23.3 | 45.2 |
| | PTv3 [54] | Overlap Detection (*ours*) | **24.8** | **15.4** | **50.2** |

**Ablation on Overlap-aware Pretraining.** The first block of Tab. 4 compares four pretraining strategies for our flow-based assembly model on BreakingBad–Everyday [12]. The first two encoders (MLP and PTv3 without pre-training) are trained jointly with the flow model. The last two encoders are PTv3 pretrained on instance segmentation and our overlap-aware prediction tasks, respectively. Their pretrained weights are frozen during flow model training. We find that PTv3 is a more powerful feature encoder compared to the MLP, and pretraining on instance segmentation can already extract useful features for pose estimation, while our proposed overlap-aware pretraining leads to the best accuracy. We hypothesize that, although the segmentation backbone provides strong semantic features, only our overlap prediction task explicitly encourages the encoder to learn fine-grained part interactions and pre-assembly cues, critical for precise assembly and registration.

To further compare against autoencoder-based pretraining, we substitute our encoder with Point-BERT [62]'s encoder, which is pre-trained on ShapeNet [63], a superset of PartNet. We then train the flow model on PartNet under the same protocol. As reported in Tab. 4, Point-BERT encoder reduces the Part Accuracy from 50.2% to 45.2%, and worsens the accuracy in both TE and RE. We attribute this performance drop to: (i) Point-BERT is pre-trained for masked point cloud completion, which does not explicitly encourage the encoder to capture inter-part relations (e.g., contacts) that our overlap-aware pretraining targets; and (ii) PointBERT's default 64-group tokenization aggregates points into relatively coarse groups, losing fine-grained geometric details for accurate pose estimation.

**Shape Prior Learning.** To probe whether our model learns the shape priors of assembled objects better than pose-vector methods, we construct a cylindrical toy dataset. We train using a single part partition scheme and then evaluate on the same cylinder shapes but with different partition schemes.

Specifically, we generated 6,000 training cylinders with heights and diameters uniformly sampled from $[0.2, 1.0]$ m, each cut into two parts by a horizontal plane at a random height. For testing, 600 new cylinders were cut under three schemes: (i) *Horizontal* (in-distribution, ID), (ii) *Axial*: through the central axis at a random orientation (out-of-distribution, OOD), and (iii) *Random*: through random 3D plane (OOD). As shown in Tab. 5, our method achieved part accuracies of 100.0%, 97.0%, and 97.8% on three partition schemes, respectively. While GARF has comparable performance on the ID scheme, it degrades on two OOD schemes, verifying that our model learns a transferable shape prior over the whole assembled object rather than overfitting to the part partition-specific patterns.

Table 5: **Part Accuracy [%] on Testing Part Schemes**. Our model shows much less drop on OOD partitions.

| Partition Scheme | GARF | *Ours* |
|---|---|---|
| Horizontal (ID) | 99.5 | **100.0** |
| Axial (OOD) | 89.5 | **97.0** |
| Random (OOD) | 87.5 | **97.8** |
| **All** | 92.1 | **98.3** |

Table 6: **Zero-shot Evaluation on the Unseen FRACTURA Testset.** We report Part Accuracy (%) for GARF and ours and include the supervised performance for reference.

| Setting | Method | Leg | Hip | Rib | Vertebra | Pig Bones | All |
|---|---|---|---|---|---|---|---|
| *Supervised* | GARF | *89.7* | *80.8* | *74.8* | *60.8* | *79.0* | *77.3* |
| Zero-shot | GARF | 70.5 | **72.8** | 62.9 | 37.7 | 53.4 | 57.7 |
| | *Ours* | **79.9** | 63.4 | **76.2** | **42.0** | **63.2** | **64.4** |

**Generalization to Unseen Dataset.** We evaluated the out-of-domain generalization of our model by performing *zero-shot* tests on unseen bone fracture on the test split of the FRACTURA dataset [11], covering human bones (Leg, Hip, Vertebra, Rib) and pig bones. We report Part Accuracy for our method and GARF [11] in Tab. 6. Our model achieves strong zero-shot performance, surpassing GARF in most categories, especially Leg and Rib, and higher overall accuracy. While there remains a gap to fully supervised training, we expect that pretraining and/or fine-tuning on medical datasets will further improve our model by instilling bone-specific shape priors and fracture geometry cues.

## 5 Conclusion

We introduce Rectified Point Flow, a unified flow-based framework for point cloud pose estimation across registration and assembly tasks. By modeling part poses as velocity fields, it captures fine geometry, handles symmetries and part interchangeability, and scales to varied part counts via joint training on 100K shapes. Our two-stage pipeline—overlap-aware encoding and rectified flow training—achieves state-of-the-art results on six benchmarks. Our work opens up new directions for robotic manipulation and assembly by enabling precise, symmetry-aware motion planning.

**Limitations and Future Work.** While our experiments focus on object-centric point clouds, real-world scenarios often involve cluttered scenes and partial observations. Moreover, while our model can generate multiple plausible assemblies, some of these may not be mechanically functional; see Fig. 7 (first column). Also, our model cannot handle objects that exceed a certain geometric complexity; see Fig. 7 (second column). Another limitation arises from the number of points our model can handle, which may restrict its usage on large-scale objects. Future work will extend Rectified Point Flow to robustly handle occlusion, support scene-level and multi-body registration, incorporate object-function reasoning, and scale to objects with larger point clouds.

**Broader Impact.** Rectified Point Flow makes it easier to build reliable 3D alignment and assembly systems straight from raw scans, benefiting robotics, digital manufacturing, AR, and heritage reconstruction. Given its performance and speed, it reduces the barrier for applying 3D part reasoning in resource-constrained settings. However, the model can still produce incorrect, hallucinated, or nonfunctional assemblies. For safety, further work on assembly verification and assembly error recovery will be essential.

## Acknowledgments and Disclosure of Funding

Tao Sun is supported by the Stanford Graduate Fellowship (SGF). Liyuan Zhu is partially supported by SPIRE Stanford Student Impact Fund Grant. Shuran Song is supported by the NSF Award #2037101. We appreciate the authors of GARF [11] for providing the FRACTURA dataset. We also thank Stanford Marlowe Cluster [64] for providing GPU resources.

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

# Rectified Point Flow: Generic Point Cloud Pose Estimation
## Supplementary Material

In this supplementary material, we provide the following:

- **Model Details** (Sec. A): Description of the DiT architecture and positional encoding scheme.
- **Additional Evaluation** (Sec. B):
    - Runtime analysis.
    - Evaluation on the preservation of rigidity at the part level.
    - Comparison against category-specific assembly models on PartNet and IKEA-Manual.
    - Analysis of different generative formulations,
    - Evaluation of the anchor-free version of our model.
- **Randomness in Assembly Generation** (Sec. C): Investigation of the assemblies generated through noise sampling and linear interpolation in the noise space.
- **Generalization Ability** (Sec. D): Qualitative results on unseen assemblies with same- or cross-category parts to test the model's generalization ability.
- **Proof of Theorem 1** (Sec. E): Formal definition of the assembly symmetry group $\mathcal{G}$ and complete proof of the flow invariance under the group $\mathcal{G}$.
- **Generalization Bounds** (Sec. F): Derivation of the generalization risk guarantees and comparison with that of existing 6-DoF methods.

## A  Model Details

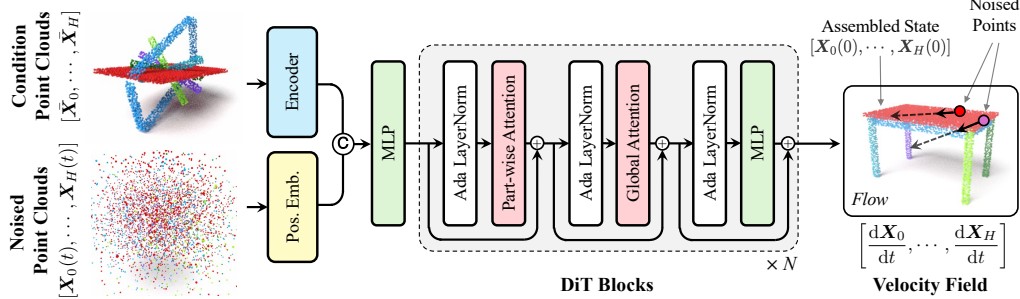

Figure 8: **Details of the DiT Block.** Our flow model consists of an Encoder and a position embedding (Pos. Emb.), and sequential DiT blocks ($N = 6$). Each block comprises Part-wise Attention, Global Attention, MLP, and AdaLayerNorm layers.

**DiT Architecture.**    Our flow model consists of 6 sequential DiT [55] blocks, each with a hidden dimension of 512. For the multi-head self-attention in the DiT block, we set the number of attention heads to 8, resulting in a head dimension of 64. As illustrated in Figure 8, inspired by [73], we apply separated *Part-wise Attention* and *Global Attention* operations in each DiT block to capture both intra-part and inter-part context:

- **Part-wise Attention:** Points within each part independently undergo a self-attention operation, improving the model's ability to capture local geometric structures.
- **Global Attention:** Subsequently, global self-attention operation is applied to all points across parts, facilitating inter-part information exchange.

As discussed in Sec. 4, we apply RMS normalization individually to the query and key features in each attention head before both attention operations to enhance numerical stability during training. Additionally, every DiT block employs AdaLayerNorm, a layer normalization whose scaling and shifting parameters are modulated by the time step $t$, following [55].

**Positional Encoding.** We adopt a multi-frequency Fourier feature mapping [70], to encode spatial information in both the condition and noised point clouds. For the $j$-th point in the $i$-th part in the condition point cloud, $\mathbf{x}_{i,j} \in \mathbf{X}$, we construct a 10-dimensional vector, which comprises:

- The 3D absolute coordinates of $\mathbf{x}_{i,j}$.
- The 3D surface normal $\boldsymbol{n}_{i,j}$ at that point $\mathbf{x}_{i,j}$ in the condition point cloud.
- The 3D absolute coordinates of the noised point cloud $\mathbf{X}(t)$ at the index $(i,j)$.
- The scalar part index $i$.

Each of these vectors is mapped through sinusoidal embeddings at multiple frequencies and then concatenated with the point-wise feature output of encoder $F$.

**Inference.** At inference time, we recover the assembled point cloud of each part by numerically integrating the predicted velocity fields $\mathbf{V}(t, \{\mathbf{X}_i(t)\}_{i\in\Omega} \mid \mathbf{X})$ from $t = 1$ to $t = 0$. In practice, we perform $K$ uniform Euler steps as,

$$\hat{\mathbf{X}}(t - \Delta t) = \hat{\mathbf{X}}(t) - \mathbf{V}(t, \{\mathbf{X}_i(t)\}_{i\in\Omega} \mid \mathbf{X})\Delta t, \quad \text{where } \Delta t = 1/K.$$

After $K$ iterations, the resulting $\hat{\mathbf{X}}(0)$ approximates the point clouds of all parts in the assembled state. For all evaluations, we set $K = 20$.

# B   Additional Evaluation

**Runtime Analysis.** The number of sampling steps of the flow model is an important factor that affects both accuracy and runtime. In Tab. 7, we vary the sampling steps and report the Part Accuracy, Chamfer Distance, and the runtime per sample in PartNet-Assembly, measured on a single RTX 4090 GPU. Increasing steps consistently improves accuracy and geometric precision, with diminishing returns beyond 20 steps. We use $K = 20$ sampling steps for all evaluations in the paper. In this setting, our model achieves 4.3 samples per second, making it practical for robotic assembly and localization tasks that require frequent online inference.

Table 7: **Trade-offs of Sampling Steps between Accuracy and Runtime.** Metrics and runtime evaluated on PartNet-Assembly dataset with a single NVIDIA RTX 4090 GPU.

| Sampling steps | 1 | 2 | 5 | 10 | 20 | 50 |
|---|---|---|---|---|---|---|
| Part Accuracy [%] ↑ | 25.2 | 38.1 | 46.7 | 52.1 | 53.9 | 54.6 |
| Chamfer Distance [cm] ↓ | 3.23 | 1.70 | 0.90 | 0.75 | 0.73 | 0.71 |
| Runtime / sample [s] ↓ | 0.072 | 0.081 | 0.108 | 0.148 | 0.232 | 0.483 |

**Part-level Rigidity Preservation.** As a dense point map prediction framework, Rectified Point Flow is not explicitly trained to preserve the rigidity of each part. To quantify how well it preserves the rigidity of the parts, we first align each predicted part $\hat{\mathbf{X}}_i(0)$ with the part in assembled state $\mathbf{X}_i(0)$ using the Kabsch algorithm.

We then measure two rigidity preservation errors using (1) the Root Mean Square Error (RMSE) over all points and (2) the Overlap Ratio (OR) over all points at varying thresholds $\tau \in \{0.1, 0.2, 0.5, 1, 2\}$ cm. Specifically, for part $i$ with $M_i$ points, we compute

$$\text{RMSE} = \sqrt{\frac{1}{M_i} \sum_{j=1}^{M_i} \left\| T_i' \hat{\mathbf{x}}_{i,j}(0) - \mathbf{x}_{i,j}(0) \right\|^2} \quad \text{and}$$

$$\text{OR}(\tau) = \frac{1}{M_i} \left| \{j \mid \| T_i' \hat{\mathbf{x}}_{i,j}(0) - \mathbf{x}_{i,j}(0) \| < \tau\} \right|.$$

Here, $T_i' \in \text{SE}(3)$ denotes the optimal rigid transform returned by Kabsch; $\mathbf{x}_{i,j}$ and $\hat{\mathbf{x}}_{i,j}$ denote the $j$-th point on $\mathbf{X}_i(0)$ and on $\hat{\mathbf{X}}_i(0)$, respectively. Because each part is first rigidly aligned to the ground-truth assembled state, these metrics intentionally ignore pose errors, and only measure the shape difference between the predicted and ground truth point parts. To factor in the variations in

Table 8: **Part-level Rigidity Preservation Evaluation.** Rectified Point Flow demonstrates low shape discrepancy between condition and predicted part point clouds, measured by the Root Mean Square Error (RMSE), Relative RMSE, and Overlap Ratios (ORs) across datasets. $D$ represents the average object scale of each dataset. (*Abbr*: BreakingBad-E = BreakingBad-Everyday; PartNet-A = PartNet-Assembly; IKEA-M = IKEA-Manual.)

| Metric | | Shape Assembly | | | | Pairwise Registration | |
|---|---|---|---|---|---|---|---|
| | | BreakingBad-E | TwoByTwo | PartNet-A | IKEA-M | TUD-L | ModelNet-40 |
| Object Scale $D$ | [cm] – | 52.1 | 107.7 | 89.0 | 61.4 | 40.8 | 70.0 |
| RMSE | [cm] ↓ | 0.76 | 2.46 | 1.04 | 0.66 | 0.16 | 0.30 |
| Relative RMSE | [%] ↓ | 1.5 | 2.3 | 1.2 | 1.1 | 0.4 | 0.4 |
| OR ($\tau = 0.1$ cm) | [%] ↑ | 52.3 | 63.8 | 33.1 | 46.7 | 96.9 | 95.0 |
| OR ($\tau = 0.2$ cm) | [%] ↑ | 61.7 | 70.8 | 48.6 | 57.7 | 97.1 | 96.0 |
| OR ($\tau = 0.5$ cm) | [%] ↑ | 74.9 | 76.8 | 66.8 | 69.8 | 97.4 | 96.3 |
| OR ($\tau = 1$ cm) | [%] ↑ | 81.4 | 78.7 | 77.9 | 81.0 | 97.7 | 96.6 |
| OR ($\tau = 2$ cm) | [%] ↑ | 89.5 | 81.9 | 87.4 | 92.0 | 98.2 | 97.1 |

object size across datasets, we compute the average scale of an object, denoted by $D$, as twice the average distance from the object's center of gravity to all its points. Then, we define the Relative RMSE as RMSE / $D$, *i.e.*, the RMSE normalized by the average object scale. We report these metrics averaged for all parts in each dataset in Tab. 8.

For the **pairwise registration** task, Rectified Point Flow demonstrates strong rigidity preservation. On TUD-L, we obtain a Relative RMSE of 0.4% and ORs above 96.9% even at the strictest $\tau = 0.1$ cm threshold; on ModelNet-40, we achieve the same Relative RMSE of 0.4% with similar high ORs above 95.0%. Specifically, on TUD-L we record ORs of 96.9% ($\tau = 0.1$ cm), 97.1% ($\tau = 0.2$ cm), 97.4% ($\tau = 0.5$ cm), 97.7% ($\tau = 1$ cm) and 98.2% ($\tau = 2$ cm); on ModelNet-40 the corresponding ORs are 95.0%, 96.0%, 96.3%, 96.6% and 97.1%, demonstrating consistently strong rigidity preservation.

In the more challenging **shape assembly** task, rigidity errors remain low. Across the four datasets, the Relative RMSE ranges from 1.1% to 2.3%. At a strict threshold of $\tau = 0.1$ cm, overlap ratios (ORs) span 33.1 % (PartNet-Assembly) up to 63.8 % (TwoByTwo); By $\tau = 1$ cm, the ORs exceed 77.9% in the four datasets (77.9%-81.4%), increasing further to 81.9%-92.0% in the more relaxed $\tau = 2$ cm. The highest Relative RMSE and lower averaged ORs are observed in TwoByTwo, probably due to its limited training samples and lower shape similarity to other datasets, and the fact that TwoByTwo has the largest overall object scale of 107.7 cm among all datasets. In contrast, IKEA-Manual, despite having fewer training samples, benefits from shared priors in furniture objects in joint training, delivering the lowest RMSE and high ORs at all thresholds. These results demonstrate robust rigidity preservation of Rectified Point Flow even in complex shape assembly scenarios.

Furthermore, please note that the subsequent pose recovery stage in Rectified Point Flow further refines part poses via an SVD-based global optimization, which fits optimal poses under noises. Overall, we empirically confirm that Rectified Point Flow generates point clouds that reliably respect the rigid structure of the conditioning parts.

**Comparison with Category-specific Models.** We compare against category-specific point-cloud assembly methods in Tab. 9. All baselines are trained separately for each category, and the category labels are assumed to be known at inference time. RGL-Net [66] additionally assumes a top-to-bottom ordering of the input parts. In contrast, Rectified Point Flow is *class-agnostic* and performs inference without any class label or part ordering. We evaluated both shape Chamfer Distance (CD) and Part Accuracy (PA) in PartNet-Assembly and IKEA-Manual, following the protocol of Huang *et al.* [47].

Without category or ordering assumptions like the baseline methods, our joint model still achieves the lowest CD and matches or exceeds the PA of category-specific baselines optimized for each category (chair, table, lamp). In particular, we observe a relative improvement of 110.2% on Lamps PA over the strongest baseline. In IKEA-Manual, we observe that all category-specific models collapse to PA $\leq 6.9\%$ for the Chair category. We hypothesize that the baselines' architecture and hyperparameter are largely tailored to PartNet. In contrast, our joint model achieves 29.9% PA for the Chair category

Table 9: **Comparison with Category-specific Models.** We report Shape Chamfer Distance (CD) and Part Accuracy (PA) on the PartNet-Assembly and IKEA-Manual. All baselines are trained per category, whereas Rectified Point Flow is trained over all categories. (*RGL-Net additionally requires a top-to-bottom part ordering.)

| Method | Known Category | PartNet-Assembly | | | | | | | | IKEA–Manual [17] | | | |
| | | Chair | | Table | | Lamp | | All | | Chair | | All | |
| | | CD↓ [cm] | PA↑ [%] | CD↓ [cm] | PA↑ [%] | CD↓ [cm] | PA↑ [%] | CD↓ [cm] | PA↑ [%] | CD↓ [cm] | PA↑ [%] | CD↓ [cm] | PA↑ [%] |
|---|---|---|---|---|---|---|---|---|---|---|---|---|---|
| B-LSTM [65] | ✓ | 1.31 | 21.8 | 1.25 | 28.6 | 0.77 | 20.8 | – | – | 1.81 | 3.5 | – | – |
| B-Global [65] | ✓ | 1.46 | 15.7 | 1.12 | 15.4 | 0.79 | 22.6 | – | – | 1.95 | 0.9 | – | – |
| RGL-Net* [66] | ✓ | 0.87 | **49.2** | 0.48 | **54.2** | 0.72 | 37.6 | – | – | 5.08 | 4.0 | – | – |
| Huang *et al.* [65] | ✓ | 0.91 | 39.0 | 0.50 | 49.5 | 0.93 | 33.3 | – | – | 1.51 | 6.9 | – | – |
| *Ours* (*Joint*) | ✗ | **0.71** | 44.1 | **0.36** | 49.4 | **0.49** | **70.0** | **0.48** | **53.9** | 1.49 | **29.9** | **0.48** | **33.2** |

Table 10: **Generative Formulation Comparison.** We compare Rectified Flow (RF) with Denoising Diffusion Probabilistic Model (DDPM) in our method, with both using the same DiT architecture and pretrained encoder. RF achieves superior performance on Rotation Error (RE) and Translation Error (TE) across all datasets. (*Abbr*: BreakingBad-E = BreakingBad-Everyday; PartNet-A = PartNet-Assembly; IKEA-M = IKEA-Manual.)

| Metric | Generative Formulation | Shape Assembly | | | | Pairwise Registration | |
| | | BreakingBad-E | TwoByTwo | PartNet-A | IKEA-M | TUD-L | ModelNet-40 |
|---|---|---|---|---|---|---|---|
| RE [deg]↓ | DDPM | 13.0 | 17.2 | 29.5 | 21.4 | 2.6 | 3.4 |
| | RF | **7.4** | **13.2** | **21.8** | **10.8** | **1.4** | **0.9** |
| TE [cm]↓ | DDPM | 3.5 | 10.1 | 21.3 | 19.2 | 0.5 | 0.7 |
| | RF | **2.0** | **3.0** | **14.8** | **17.2** | **0.3** | **0.2** |

and 33.2% PA for all categories, over 4 times higher than any baselines. Those observations confirm that our category-agnostic cross-dataset training improves the learning of shared geometric priors far beyond any single category or dataset.

**Ablation on Generative Formulation.** As an alternative to the generative formulation of Rectified Flow (RF) in our method, we also evaluate a Denoising Diffusion Probabilistic Model (DDPM) [68] using an identical DiT architecture and the pre-trained encoder. In this setup, the forward noising process employs constant variances ($\beta$) that increase linearly from $10^{-4}$ to 0.02 over $T = 1000$ timesteps. As shown in Tab. 10, the RF-based model consistently outperforms the DDPM variant on both shape assembly and pairwise registration tasks, with 35.3% lower rotation error (RE) and 11.63% lower translation error (TE). This result is in line with the findings of GARF [11]. We hypothesize that the straight-line flow in RF reduces the learning difficulty in our tasks. DDPM's frequency-based generation—which works well for images—may not be as effective as RF for 3D point cloud synthesis in Euclidean space.

**Anchor-free Models.** In the anchor-free setting, we do *not* fix the anchor's pose. Instead, during training, we treat the anchor exactly like any other part in the conditioning: its point cloud is centered to its own CoM frame and then randomly rotated. The model, therefore, never receives the true anchor pose from condition. The flow target is defined in the anchor frame: we put the completed assembly point cloud in the anchor's frame and then re-center the whole assembled point cloud. At inference, we first estimate the rigid transform between the predicted anchor and the ground-truth anchor point cloud via ICP, and then apply this transform to *all* predicted parts. Metrics (RE, TE, and Part Acc.) are computed after this anchor alignment.

Across shape assembly datasets (Tab. 11) and pairwise registration (Tab. 12), **anchor-free** (average) underperforms **anchor-fixed** as expected due to (i) error propagation from anchor misalignment and (ii) ambiguity induced by symmetric anchor parts. However, despite the lower absolute performance, our anchor-free model preserves the same relative ordering among baselines on all shape-assembly datasets; the only exception is GARF, which is evaluated in the anchor-fixed protocol. In particular,

Table 11: **Evaluation of the Anchor-free Model on Shape Assembly Datasets.**

| Dataset | Anchor-fixed | | | Anchor-free (Average) | | | Anchor-free (Best-of-3) | | |
|---|---|---|---|---|---|---|---|---|---|
| | RE ↓ [deg] | TE ↓ [cm] | PA ↑ [%] | RE ↓ [deg] | TE ↓ [cm] | PA ↑ [%] | RE ↓ [deg] | TE ↓ [cm] | PA ↑ [%] |
| BreakingBad-E [12] | 7.4 | 2.0 | 93.5 | 17.4 | 8.0 | 90.2 | 13.0 | 5.9 | 92.2 |
| TwoByTwo [16] | 13.2 | 3.0 | – | 15.2 | 24.2 | – | 9.0 | 16.7 | – |
| PartNet-Assembly [15] | 21.8 | 14.8 | 53.9 | 47.3 | 40.5 | 45.3 | 38.2 | 32.9 | 54.3 |
| IKEA-Manual [17] | 20.7 | 24.7 | 33.2 | 54.7 | 51.5 | 19.5 | 39.0 | 40.5 | 28.6 |

Table 12: **Evaluation of the Anchor-free Model on Pairwise Registration Datasets.**

| Setting | ModelNet-40 [19] | | TUD-L [18] | |
|---|---|---|---|---|
| | RE ↓ [deg] | TE ↓ [unit] | Recall @ 5° ↑ [%] | Recall @ 1cm ↑ [%] |
| Anchor-fixed | 0.93 | 0.002 | 97.7 | 99.0 |
| Anchor-free (Average) | 2.05 | 0.012 | 96.6 | 96.3 |
| Anchor-free (Best-of-3) | 1.18 | 0.007 | 97.3 | 97.3 |

our model significantly improves anchor-free SOTA on BreakingBad-Everyday's Part Accuracy from 76.2% (PuzzleFussion++ [25]) to 90.2%. For pairwise registration, our model is also achieving the lowest TE on ModelNet and the highest Recall@5° on TUD-L among baselines, while remain competitive on other metrics.

Furthermore, we observe that the **anchor-free** (best-of-3) evaluation, which selects the best prediction out of 3 different random seeds per sample, largely narrows the gap to **anchor-fixed**. This indicates that a small stochastic budget may find a more reliable anchor-free prediction. The effect is most pronounced on datasets with higher part count and weaker geometric cues (e.g., PartNet-Assembly and IKEA-Manual), where anchor ambiguity is stronger and small anchor errors propagate severely.

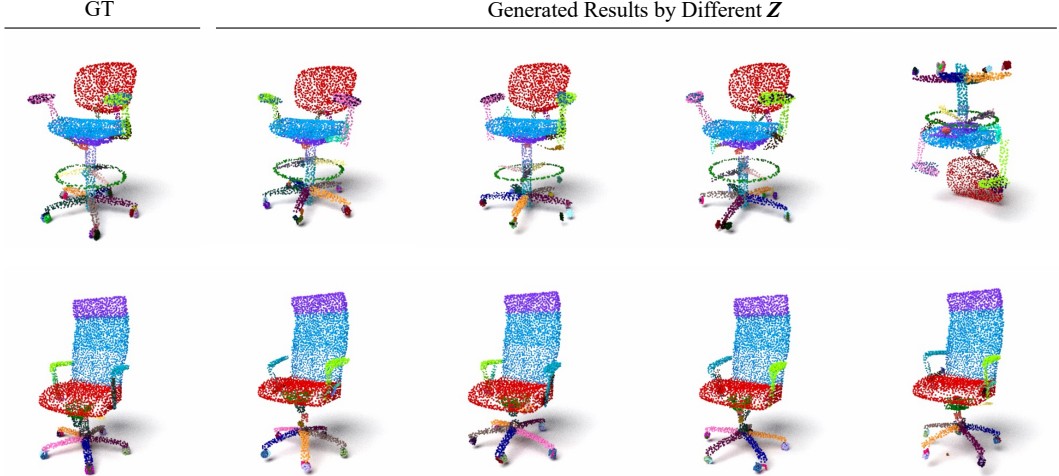

Figure 9: **Sampling in Noise Space.** For each fixed condition input point clouds, we sample four independent Gaussian noise vectors to generate distinct assembly outputs (shown in columns 2–5). While all samples preserve the object's structure, they show meaningful variation in part placement, orientation, and overall geometry, particularly for symmetric parts (*e.g.*, armrests and chair bases). For comparison, the first column shows the ground-truth assemblies.

# C   Randomness in Assembly Generation

**Diversity via Noise Sampling.**   To evaluate the diversity of assembly configurations generated by Rectified Point Flow, we sample the Gaussian noise vector $\boldsymbol{Z}$ multiple times for the same conditional (unposed) point cloud inputs. At inference time, we set $\boldsymbol{X}(1) = \boldsymbol{Z}$ and run the model to obtain prediction $\hat{\boldsymbol{X}}(0)$. In Figure 9, each row corresponds to a single final assembly: the first column shows the ground-truth assembly, and the next four columns display outputs produced by four different Gaussian noises. All generated assemblies preserve the part structure, yet exhibit meaningful variations in the parts' placement and orientation, and overall geometry of the object. As expected, the model produces diverse configurations for symmetric or interchangeable parts, such as the armrests and the chair base. This shows that Rectified Point Flow effectively captures a diverse conditional distribution of valid assemblies.

**Linear Interpolation in Noise Space.**   We illustrate Rectified Point Flow's learned mapping from random Gaussian noise to plausible assembly configurations. In Figure 10, each row uses the same condition (unposed) point cloud, with the left and right columns showing the outputs of two randomly sampled noise vectors $\boldsymbol{Z}_0$ and $\boldsymbol{Z}_1$, respectively. The three columns in between display results generated by $\boldsymbol{Z}(s)$ which linearly interpolates between $\boldsymbol{Z}_0$ and $\boldsymbol{Z}_1$ in noise space, *i.e.*,

$$\boldsymbol{Z}(s) := (1 - s)\boldsymbol{Z}_0 + s\boldsymbol{Z}_1, \quad \text{where } s \in \{0.25, 0.5, 0.75\}. \tag{7}$$

At each interpolation step $s$, we run inference with $\boldsymbol{X}(1) = \boldsymbol{Z}(s)$. As $s$ increases, the predicted shapes smoothly morph from the configuration induced by $\boldsymbol{Z}_0$ toward that of $\boldsymbol{Z}_1$. As shown in the first 2 rows in Figure 10, we observe smooth transitions among interchangeable parts in both examples. The 2 bottom rows in Figure 10 visualize the transitions in the overall structure of objects. In the table example, we observe a gradual reduction in overall height, a lowering of the horizontal beams, and a more centralized positioning where the four legs meet. In the shelf example, the transformation

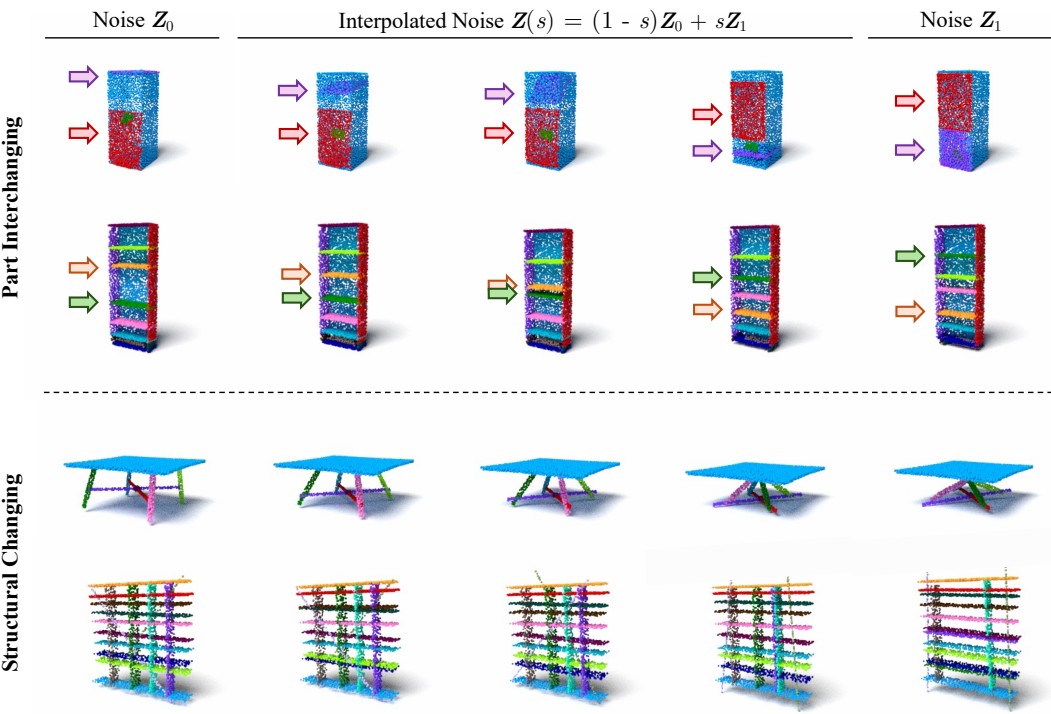

Figure 10: **Linear Interpolation in Noise Space.** For different objects in each row, we fix the same conditional input and decode two independently sampled Gaussian noise vectors, $\boldsymbol{Z}_0$ (leftmost) and $\boldsymbol{Z}_1$ (rightmost), into plausible part configurations. The three center columns show outputs from the linearly interpolated noises between $\boldsymbol{Z}_0$ and $\boldsymbol{Z}_1$. We observe a continuous, semantically meaningful mapping from Gaussian noise to valid assemblies.

| Object 1 | Object 2 | Generated Results |
|----------|----------|-------------------|

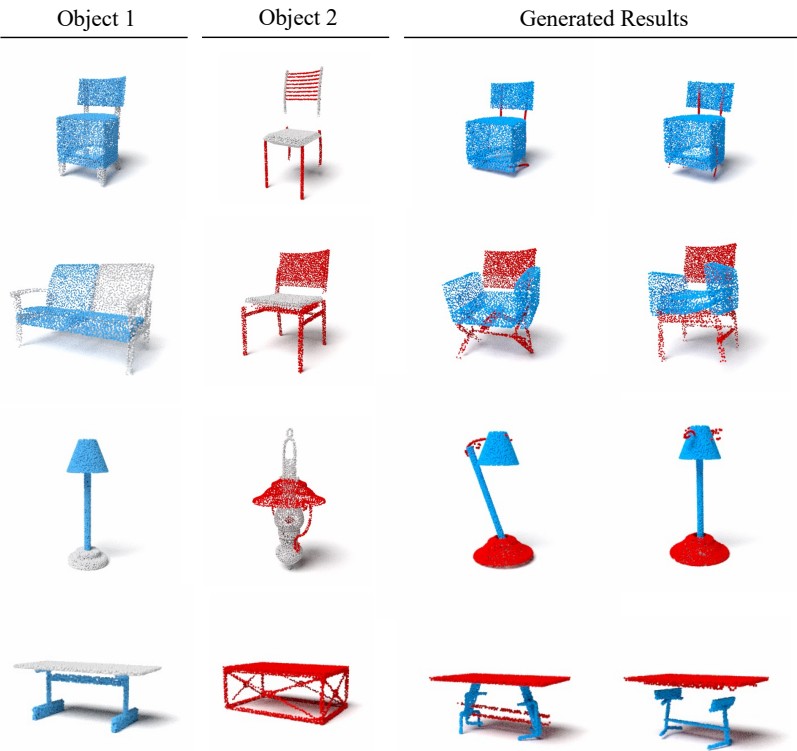

Figure 11: **Generalization to Unseen Assemblies Within the Same Category**: We select parts from two objects of the same category in the PartNet-Assembly test set. Parts from Object 1 are shown in blue, and parts from Object 2 in red; unselected parts are shown in gray. The results demonstrate that the model comprehends the underlying geometric structure of the category and can re-target parts to construct the final shape.

is more drastic: two vertical boards become horizontal and two diagonal cables are rearranged to a new vertical configuration. The above transitions across various assemblies confirm that Rectified Point Flow learns a continuous mapping from Gaussian noise to a semantically meaningful geometry space. Note that most of the interpolated configurations are physically plausible assemblies, creating functional objects that can stand in real-world.

## D  Generalization Ability

We test the generalization ability of our model for novel assemblies under two different settings: between objects from the same (in-category) and different (cross-category) categories. Given two objects in PartNet-Assembly, we select certain parts from each of them as the input to Rectified Point Flow to test if the model can generate novel and plausible assemblies.

**In-category Test.**   As shown in Fig. 11, parts selected from Object 1 are rendered in blue and those from Object 2 in red. Our model then synthesizes novel assemblies that blend and reconfigure these parts in a coherent and category-consistent manner. For example, in the chair category (first two rows), the model successfully retains a functional and plausible seat-back-leg structure while creatively mixing parts. In the lamp category (third row), even though the base and shade style differ significantly between objects, generated results exhibit sensible combinations that maintain structural integrity. Similarly, in the table category (last row), our method combines parts from a flat-top table and lattice-style base to produce hybrid yet coherent table designs.

**Cross-category Test.**   Fig. 12 highlights Rectified Point Flow's ability to generalize to unseen part combinations across categories. This is a particularly challenging test, since such part combination

| Object 1 | Object 2 | Generated Results |
|---|---|---|

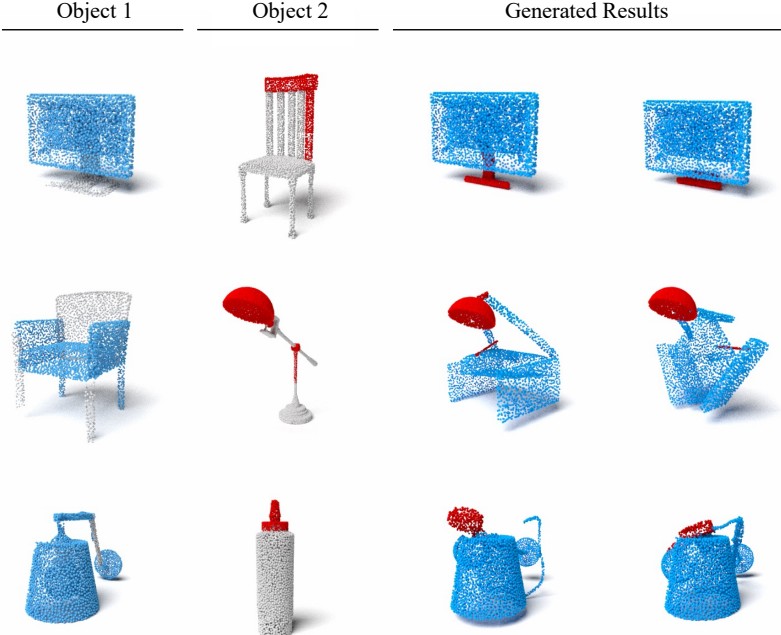

Figure 12: **Generalization to Unseen Assemblies Across Categories**: We select parts from two objects of different categories in the PartNet-Assembly test set. Parts from Object 1 are shown in blue, and parts from Object 2 in red; unselected parts are shown in gray. The results demonstrate that the model can reason about part compositionality and re-target parts to construct a plausible final shape even if some of them originate in completely different objects.

may not even be possible to be assembled into a meaningful object. Nevertheless, our method still demonstrates a certain degree of generalization. We show two input objects from different categories, for example, a monitor and a chair, a chair and a lamp, or a wall sconce and a spray bottle. The results on the right demonstrate that our model can reconfigure these parts into plausible new assemblies, preserving geometric coherence. This suggests that the model has learned a strong understanding of part relationship, allowing it to reason about compositionality even across category boundaries.

## E   Proof of Theorem 1

A key advantage of Rectified Point Flow is that it learns both rotational symmetries of individual parts and the interchangeability of a set of identical parts, without any labels of symmetry parts. Below, we first formally define an *assembly symmetry group* $\mathcal{G}$ that characterizes the symmetry and interchangeability of the parts in the multi-part point cloud.

**Definition 1** (Assembly symmetry group). *For each part $i \in \Omega$, let $G_i \subseteq \mathrm{SO}(3)$ be the (finite) stabilizer of its assembled shape, i.e., $R\boldsymbol{X}_i(0) = \boldsymbol{X}_i(0)$ for all $R \in G_i$. Let $S \subseteq \mathfrak{S}_{|\Omega|}$ be the set of permutations that only permute indices of identical parts. We define the assembly symmetry group as the semidirect product*

$$\mathcal{G} = \big( G_1 \times \cdots \times G_{|\Omega|} \big) \rtimes S. \tag{8}$$

*A group element $g = (R_1, \ldots, R_{|\Omega|}, \sigma) \in \mathcal{G}$ acts on every realization of the Rectified Point Flow by $g(\boldsymbol{X}_i(t)) := R_i \, \boldsymbol{X}_{\sigma^{-1}(i)}(t)$, and on network outputs of the $i$-th part (denoted as $\boldsymbol{V}_i$) by $g(\boldsymbol{V}_i(t, g(\boldsymbol{X}))) := R_i \, \boldsymbol{V}_{\sigma^{-1}(i)}(t, g(\boldsymbol{X})).$*

Now, we show the following result that a single point's flow distribution is invariant under any $g \in \mathcal{G}$.

**Lemma 1** ($\mathcal{G}$-invariance of the flow distribution). *For every element $g \in \mathcal{G}$ and a given multi-part point cloud $\boldsymbol{X}$, we sample a flow realization:*

$$\mathbf{x}(t) = t\mathbf{x}(1) + (1 - t)\mathbf{x}(0), \quad where \; \mathbf{x}(1) \sim \mathcal{N}(\mathbf{0}, \boldsymbol{I}), \mathbf{x}(0) \sim \boldsymbol{X}.$$

*then, we have*

$$p\big(\{g \cdot \mathbf{x}(t)\}_{t \in [0,1]}\big) = p(\{\mathbf{x}(t)\}_{t \in [0,1]}).$$

*Proof.* Recall that, in Rectified Point Flow, a flow of a single point is $\mathbf{x}(t) := (1-t)\mathbf{x}(0) + t\mathbf{x}(1)$, where $\mathbf{x}(0) \sim \boldsymbol{X}$ is drawn uniformly from the assembled shape and $\mathbf{x}(1) \sim \mathcal{N}(\mathbf{0}, \mathbf{I})$. Because the end–points of the linear interpolation are sampled independently, the PDF of the path distribution factorizes as

$$p\big(\{\mathbf{x}(t)\}_{t \in [0,1]}\big) = p\big(\mathbf{x}(1)\big)p\big(\mathbf{x}(0)\big), \tag{9}$$

which indicates the randomness resides by the states $t = 0$ and $t = 1$ only. Because the perturbation $\mathbf{x}(1) \sim \mathcal{N}(\mathbf{0}, \mathbf{I})$ is isotropic, $p(\mathbf{x}(1))$ is invariant under *every* rotation $R \in \mathrm{SO}(3)$. For $p(\mathbf{x}(0))$ we distinguish two cases:

- *Rotational symmetry:* If $R \in G_i$, then $R\boldsymbol{X}_i(0) = \boldsymbol{X}_i(0)$ point-wise, so $p(\mathbf{x}(0)) = p(R\mathbf{x}(0))$.

- *Interchangeability.* If parts $i$ and $j$ are identical, sampling first a part index with probability $p(i) = N_i/N$ and then a point uniformly inside it implies $p(\mathbf{x}(0) \in \boldsymbol{X}_i(0)) = p(\mathbf{x}(0) \in \boldsymbol{X}_j(0))$. Therefore exchanging the indices ($\sigma(i) = j, \sigma(j) = i$) leaves $p(\mathbf{x}(0))$ unchanged.

By composing the above two properties for all parts, we complete the proof. $\qquad\square$

Lemma 1 can directly lift from single points to the full multi–part flow $\{\boldsymbol{X}_i(t)\}_{i \in \Omega}$. This leads us to the Theorem 1: For every element $g \in \mathcal{G}$, we have the learning objective in Eq. 3 following $\mathcal{L}_{\mathrm{CFM}}(\boldsymbol{V}) = \mathcal{L}_{\mathrm{CFM}}(g(\boldsymbol{V}(t, \{\boldsymbol{X}_i(t)\}_{i \in \Omega}; g(\boldsymbol{X}))))$.

# F   Generalization Bounds

While the Rectified Point Flow predicts a much higher-dimensional space ($3M_i$ coordinates per part), we find that its Rademacher complexity scales exactly the same rate as the 6-DoF methods, $O(1/\sqrt{m})$, where $m$ is the number of samples in the training set.

Below, we compute their Rademacher complexities and empirical risks, respectively. Without loss of generality, we use the reconstruction error for the evaluation of poses, *i.e.*, $\ell(\hat{R}, \hat{\boldsymbol{t}}; R^\star, \boldsymbol{t}^\star) = \big\| (\hat{R} - R^\star)\boldsymbol{X}^\star + \hat{\boldsymbol{t}} - \boldsymbol{t}^\star \big\|_F$. First, we define hypothesis classes for both methods:

- Our Rectified Point Flow:

$$\mathcal{F}_i = \{C_i \mapsto \hat{\boldsymbol{X}}_i(0; \theta) \mid \theta \in \Theta\}, \quad \text{where } \hat{\boldsymbol{X}}_i(0; \theta) := \boldsymbol{X}_i(1) - \int_0^1 \boldsymbol{V}_i(t; C, \theta)\mathrm{d}t.$$

- Pose vector-based flow:

$$\mathcal{G}_i = \{C_i \mapsto (\hat{R}_i, \hat{\boldsymbol{t}}_i)_\phi \mid \phi \in \Phi\}.$$

**Rademacher Complexity of Rectified Point Flow.** With $m$ *i.i.d.* training objects $D = \{(C^{(k)}, R^{\star(k)}, \boldsymbol{t}^{\star(k)})\}_{k=1}^m$, we write the population risk $\mathcal{R}(h) = \mathbb{E}\big[\ell\big(h(C), R^\star, \boldsymbol{t}^\star\big)\big]$ and empirical risk

$$\hat{\mathcal{R}}_D(h) = \frac{1}{m} \sum_{k=1}^m \ell\big(h(C^{(k)}), R^{\star(k)}, \boldsymbol{t}^{\star(k)}\big).$$

Since our Rectified Point Flow method estimates the part pose by the Procrustes operator, *i.e.*, $(\hat{R}, \hat{\boldsymbol{t}}) = \mathrm{Pr}\big(\hat{\boldsymbol{X}}(0; \theta)\big)$, where $\mathrm{Pr} : \mathbb{R}^{3N} \to \mathrm{SE}(3)$ is the Procrustes operator, we have following Lipschitz contracting property.

**Property 1** (Lipschitz Contracting). *Let $\boldsymbol{X}^\star \in \mathbb{R}^{3N}$ be the centralized ground-truth point set of a single part, and denote $\sigma_{\min} = \sigma_{\min}((\boldsymbol{X}^\star)^\top \boldsymbol{X}^\star)$. If $\|\hat{\boldsymbol{X}}(0) - \boldsymbol{X}^\star\|_F \le \varepsilon$, the optimal Procrustes solution $(\hat{R}, \hat{\boldsymbol{t}}) = P(\hat{\boldsymbol{X}}(0))$ satisfies*

$$\|(\hat{R} - R^\star, \hat{\boldsymbol{t}} - \boldsymbol{t}^\star)\| \le \frac{\varepsilon}{\sqrt{\sigma_{\min}}}. \tag{10}$$

This property directly follows from Davis–Kahan perturbation bounds [67] for the Top-3 singular vectors. Crucially, $\sigma_{\min} = \Omega(N)^3$ for well-spread point clouds, so Pr is a $\frac{1}{\sqrt{N}}$-*Lipschitz* map.

Let $\mathfrak{R}_m(\mathcal{H})$ denote the empirical Rademacher complexity on $S$. Because composition with a $L$-Lipschitz map contracts Rademacher complexity

$$\mathfrak{R}_m(\text{Pr} \circ \mathcal{F}) \ \leq \ \frac{1}{\sqrt{N}} \, \mathfrak{R}_m(\mathcal{F}) \ \leq \ \frac{L_\Theta \sqrt{3N}}{\sqrt{N}} \, \frac{1}{\sqrt{m}} \ = \ O\Big( \frac{L_\Theta}{\sqrt{m}} \Big). \tag{11}$$

**Rademacher Complexity of 6DoF-based Methods.** For the baseline we need only regress $d = 6$ numbers, hence

$$\mathfrak{R}_m(\mathcal{G}) \ \leq \ \frac{L_\Phi \sqrt{d}}{\sqrt{m}} = O\Big( \frac{L_\Phi}{\sqrt{m}} \Big). \tag{12}$$

**Comparison of Generalization Bounds.** Applying Bartlett Theorem and using (10), we obtain, with probability at least $1 - \delta$ over the samples from $D$,

$$\mathcal{R}\big(P \circ \hat{f}\big) \ \leq \ \hat{\mathcal{R}}_D\big(P \circ \hat{f}\big) + 2\,\mathfrak{R}_m(P \circ \mathcal{F}) + 3\sqrt{\frac{\log(2/\delta)}{2m}}, \tag{FLOW}$$

$$\mathcal{R}(\hat{g}) \ \leq \ \hat{\mathcal{R}}_D(\hat{g}) + 2\,\mathfrak{R}_m(\mathcal{G}) + 3\sqrt{\frac{\log(2/\delta)}{2m}}, \tag{6DoF}$$

where $\hat{f} \in \mathcal{F}$ and $\hat{g} \in \mathcal{G}$ are the empirical-risk minimizers on $S$.

In conclusion, while Rectified Point Flow predicts a much higher-dimensional space, the contraction of the SVD stage cancels this apparent over-parameterization, producing a complexity term that scales at the same rate of $O(1/\sqrt{m})$ as the 6-DoF baseline; (FLOW)–(6DOF).

As a result, our method enjoys at least same generalization risk guarantees despite operating in an over-parameterized prediction space, while retaining the $\mathcal{G}$-invariance benefits proven in Sec. E.

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

---

[3]Here, $\Omega(\cdot)$ denote the asymptotic rate, instead of part index set.

