# OpenReview forum: "Rectified Point Flow: Generic Point Cloud Pose Estimation"
_NeurIPS.cc/2025/Conference — NeurIPS 2025 spotlight_

### Official Review · Reviewer_JGmh · 2025-06-22

**Clarity:** 4
**Significance:** 3
**Originality:** 3
**Rating:** 5
**Confidence:** 4

**Summary:**

This paper is addressing the problem of registration of point clouds. More specifically, the main focus is registration of multiple point clouds to each other resulting in an assembled object. This problem is not solving generic multi point cloud registration, but rather a specific problem, of part assembling, where the input are object parts in arbitrary poses and they can be aligned to make an object. A side problem the paper addresses, is a registration of two scans. The problem  addressed in this paper is interesting and relevant.

The paper approaches this problem using generative modelling, notably, it leverages a diffusion model. The diffusion model is basically conditioned on the unposed object parts and outputs a vector field for each point in each part, pointing towards its location in the target or assembled object. The rigid transformations are easily computed from known dense point correspondences using a SVD. For such generative models to work reasonably it is necessary to train them on a large amount of data. Authors use Objaverse dataset for initialisation, where the objects were segmented into the parts using available off the shelf methods. Later their diffusion model is trained specifically on each dataset they use in evaluation, resulting in one model per dataset. Alternatively, they trained a single model for all available datasets they use in the evaluation.

**Questions:**

The questions below are related to weaknesses, so when addressing them, please refer to corresponding weaknesses.
1) Generalization. The question is if the objects from the test split belong to the same classes as those used for training. What would be the results if the model was trained on one dataset and evaluated on the other.

2) Self-supervision. It would be good to get more details here and explain, why the authors call pre-training approach self-supervised.

3) Related work especially wrt. symmetry handling. It would be good to give an intuition about this intrinsic symmetry handling. My feeling is that the diffusion model “memorises” how to from different parts to the canonical shape, even in case of symmetries. I would be surprised if a new object with novel type of symmetry can easily be handled.

**Ethical Concerns:**

["NO or VERY MINOR ethics concerns only"]

**Final Justification:**

I appreciate the authors' rebuttal, where they addressed my concerns. Taking into account other reviews and provided answers, I am raising my score to 5.

**Paper Formatting Concerns:**

No problems identified

**Quality:**

3

**Strengths And Weaknesses:**

Strengths: The paper proposes an interesting approach for multi part registration. I find  it clearly formulated and presented. I find using a diffusion model for addressing this problem intriguing, but the formulation is nice and appealing. Experimental evaluation is substantial and contains a number of datasets and related works the paper compares too. Finally, I find the discussion useful and appreciate showing failure cases and their causes.

Weaknesses: Even though I like the approach and the formulation, I think it has several weaknesses.
1) Questionable generalisation capablities. Primarily, generative models, like diffusion models, are still bound to the datasets they are trained on. They excel in the generative tasks, like generating new shapes, interpolated between the existing one from the training. However, for the registration, they are bound to the datasets they use for training. Unlike, classical methods, e.g. keypoint matching based, which can handle any objects in any configurations, generative models, cannot be applied to completely new objects. I believe that the same applies here. The model is bound to the datasets they are trained on, even if they are trained on all datasets used in this paper. For example, the paper, follows, train/validation/test splits. This means that part of the dataset has been seen during the training of the diffusion model.

2) Self-supervised pretraining. Training the encoder and overlap head on a large dataset, as Objaverse, using cross entropy loss, where all object parts and their overlaps are known, doesn’t seem to be self-supervised, but rather supervised. Generating labels(object part IDs) and their overlaps automatically, cannot be considered as past of self-supervision. I would appreciate your comment on this.

3) Related works. The discussion about the related work is valid and well structured. However, it is hard to see how different the proposed approach is from the related diffusion models:DiffAssembly, PuzzleFusion and GARF. Claiming that others handle symmetries in an ad-hoc manner, while the proposed approach does it intrinsically is not clear.

4) Experimental evaluation. Even though the evaluation is exhaustive, I would like to see comparisons, where test objects have never been used for training. For example, a cross validation kind of evaluation would be interesting: the model is trained on all datasets except one, on which it is tested. Why there is no comparison to DiffAssembly, which seems to be quite related to the current approach? Symmetry handling example in Figure 6 is not sufficient. It would be great to have a cylindrical toy object, e.g. bottle, and show how multiple symmetric scans could be registered. Training on one set of cylindrical objects and testing on  completely different one would be beneficial to see if the method generalises in case of symmetries.

---

> ### Author Rebuttal · Authors · 2025-07-31
>
> We sincerely thank the reviewer for finding our approach "interesting" and "clearly formulated", and for describing our formulation as "nice and appealing", along with recognizing our "substantial experimental evaluation". We also greatly appreciate the constructive and insightful feedback. Below is our response to the comments.
>
> ---
>
> > **Q1**: Generalization. The question is if the objects from the test split belong to the same classes as those used for training. What would be the results if the model was trained on one dataset and evaluated on the other.
>
> We have conducted zero-shot out-of-domain evaluation and find that our method demonstrates better generalization abilities, compared to the state-of-the-art method GARF. Please refer to our response to **Reviewer oaRr** (Q1), where we report zero-shot evaluation on the **3RScan** \[A] dataset (pairwise registration under noisy real-world 3D scanning with partial observations), and to **Reviewer yLRg** (Q2) for our zero-shot evaluation on the **Fractura** \[11] dataset (shape assembly of various types of bone scans).
>
> ---
>
> > **Q2**: Self-supervision. It would be good to get more details here and explain, why the authors call pre-training approach self-supervised.
>
> Our overlap-aware pretraining uses various part segmentations, including random planar partitions (e.g., ModelNet-40). However, we agree that our Objaverse pretraining uses automatically extracted part segmentation labels by PartField, which technically falls under supervised learning or distillation. To better reflect this, we will rename this stage to "Overlap-aware Pretraining" and update Section 3.2 to clarify the distinction from fully unsupervised approaches.
>
> ---
>
> > **Q3**: Why there is no comparison to DiffAssemble, which seems to be quite related to the current approach?
>
> Thanks for the suggestion. DiffAssemble [49] applies denoising diffusion to a graph neural network (GNN) for SE(3) pose vector generation, which differs from our flow-based generative formulation. For completeness, we compared our method with DiffAssemble on the BreakingBad-Everyday, evaluating rotation error (degrees, lower is better), translation error (cm, lower is better), and part accuracy (%, higher is better):
> | Method  | Rotation Error \[deg] | Translation Error \[cm] | Part Accuracy \[%]|
> | :---: | :---: | :---: | :---: |
> | DiffAssemble | 73.3 | 14.8 | 27.5 |
> | Ours | **7.4** | **2.0** | **91.1** |
>
> As shown, our method achieves substantial improvements over DiffAssemble across all metrics. We will include these results in Table 2.
>
> ---
>
> > **Q4**: Symmetry handling example in Figure 6 is not sufficient. It would be great to have a cylindrical toy object, e.g. bottle, and show how multiple symmetric scans could be registered. Training on one set of cylindrical objects and testing on completely different one would be beneficial to see if the method generalises in case of symmetries.
> > It would be good to give an intuition about this intrinsic symmetry handling. My feeling is that the diffusion model “memorises” how to from different parts to the canonical shape, even in case of symmetries. I would be surprised if a new object with novel type of symmetry can easily be handled.
>
> Thanks for the insightful comments on symmetry generalization.
>
> We would like to emphasize that our method fundamentally handles symmetry more robustly than prior pose-vector approaches, such as DiffAssembly, PuzzleFusion, and GARF. These methods define their per-part target pose in SE(3) space, which inherently loses geometric symmetry information without explicit data augmentation. For example, consider a table with four identical cylindrical legs. Prior pose-vector approaches [11,25,41] require first annotating equivalence classes of symmetric parts (e.g., grouping legs A=B=C=D and specifying each leg’s rotational symmetry axis) and then using ad-hoc augmentations (such as swapping part locations and random rotations around those axes) during training to cover the symmetry modes. Note that those methods are originally developed for the BreakingBad dataset where there is nearly no identical fractures and only one possible way to reassemble them. Consequently, these methods often fail to generalize to other assembly tasks involving significant symmetry or part equivalence, e.g., furniture and mechanical objects.
>
> In contrast, our method requires only a single configuration (with no swapping or rotating needed) during training and can learn the full distribution over the corresponding symmetry group. Intuitively, this is because (1) our flow formulation operates in the continuous Euclidean space where the geometric symmetry information is preserved in the learning target, and (2) our point flow formulation holds equivalence over an assembly symmetry group. We provide a formal proof of this equivalence in Section E of the Supplementary Material. This property is also demonstrated in Figure 9 in the Supplementary Material, where the diversity among test samples arises mainly from symmetric part placements.
>
> Beyond generalization within the assembly symmetry group, our method also demonstrates strong generalization to **unseen** symmetry types.  To further illustrate this, we constructed a cylindrical toy dataset, as suggested. We generated 6,000 cylinders with heights uniformly sampled from \[0.2, 1.0\] m and diameters from \[0.2, 1.0\] m. Each cylinder in the training set was cut into 2 pieces only by a **horizontal** plane at a random height parallel to the cylinder's faces. For testing, we generated 600 new cylinders and applied three cut configurations:
>
> 1. **Horizontal**: by a plane at a random height parallel to the cylinder's faces, same as training,
> 2. **Axial**: by a plane through the cylinder's central axis at random orientation, and
> 3. **Random**: by an arbitrary 3D plane.
>
> Note that **Axial** and **Random** cuts have never been seen during training. The results are shown in the table below. Our method achieved high part accuracy of 100.0%, 97.0%, and 97.8% on the three configurations, respectively. While GARF shows similar results in the horizontal cut, its performance on out-of-domain (OOD) configurations is worse. This demonstrates our model’s ability to generalize to previously unseen symmetry types, rather than simply memorizing them.
>
> | Method | Horizontal | Axial (OOD) | Random (OOD) | All Cuts |
> | :---: | :---: | :---: | :---: | :---: |
> | GARF | 99.5 | 89.5 | 87.5 | 92.1 |
> | Ours | **100.0** | **97.0** | **97.8** | **98.3** |
>
> We will include this experiment and discussion of symmetry handling in the revised manuscript.

---

> > ### Comment · Reviewer_JGmh · 2025-08-04
> >
> > I appreciate the authors' rebuttal, where they addressed my concerns. I am raising my score to 5.

---

### Official Review · Reviewer_yLRg · 2025-07-02

**Clarity:** 3
**Significance:** 4
**Originality:** 4
**Rating:** 5
**Confidence:** 3

**Summary:**

This paper proposes rectified point flow, a framework that can treat point cloud registration and multi-part shape assembly in a unified manner. The key idea is to adopt the recently popular flow matching technique into the classic point cloud pose estimation problem, which remarkably accommodates both pair-wise and multiple components settings. Overall the idea is interesting, and the experimental validation suggests sufficient advantage over the competing baselines.

**Questions:**

- The reported models are mainly trained on instance-level man-made datasets. Some important applications of shape assembly come from medical tasks. I am wondering 1) the current method relies on an overlap estimation module, which requires semantic segmentation. can the framework deal with more random segments such as bone fragments due to fracture? 2) following 1), if one generates some bone fragments, which are likely unseen by the models, how well the models can generate in assembling bones? 3) if one has to train from scratch, is there any insight on the minimal requirements of training set?

- Another challenging scenario (also with strong medical background) is non-rigid shape assembly. Is there any insight on how the proposed framework can be extended to the more challenging setting?

**Ethical Concerns:**

["NO or VERY MINOR ethics concerns only"]

**Final Justification:**

I am happy with the response from the authors, and thus would maintain my initial score.

**Limitations:**

Yes.

**Paper Formatting Concerns:**

None.

**Quality:**

3

**Strengths And Weaknesses:**

Strengths:

1. Point pose estimation task is fundamental in processing 3D data. The reported method demonstrates promising results on rather challenging setting, e.g., shown in the last two columns of Fig. 4, which is potentially of great utility in various down-stream applications.

2. The proof of Thm. 1 looks reasonable to me. It provides good basis from the theoretical perspective.

3. The idea of incorporating rectified flow framework into point pose estimation is interesting, and I appreciate the authors' efforts to make this seemingly intuitive idea work in practice (e.g., the conditioning design as well as the ablation in Tab.4 suggest the efforts).

Weaknesses:

I have some questions, please refer to the *Question* part.

---

> ### Author Rebuttal · Authors · 2025-07-31
>
> We thank the reviewer for finding our idea "interesting," noting that the method "remarkably accommodates both pair-wise and multiple components settings," and "demonstrates promising results on rather challenging settings." We also appreciate the reviewer's recognition of the theoretical proof as "reasonable and providing a foundation from theoretical perspective". Please find our responses to the comments below.
>
> ---
>
> > **Q1**: The reported models are mainly trained on instance-level man-made datasets. Some important applications of shape assembly come from medical tasks. I am wondering the current method relies on an overlap estimation module, which requires semantic segmentation. can the framework deal with more random segments such as bone fragments due to fracture?
>
> Our framework is designed to accommodate various definitions of part segmentation, not only semantic ones. Specifically, we have demonstrated support for:
>
> 1. **Random fragmentation**: as in ModelNet-40, where parts are generated via random plane cuts.
> 2. **Physically simulated fracture**: as in BreakingBad, where parts are created using physical simulations of breakage when external forces are applied on object surface.
> 3. **3D scanning**: As in TUD-L, where part point clouds are acquired from real sensors, including partial occlusions and noise.
> 4. **Instance (semantic) segmentation**: As in Objaverse, PartNet, and IKEA-Manual, where parts are defined by semantic instance labels.
>
> This flexibility suggests that our method could potentially handle irregular or random segmentations, such as bone fragments from fractures, provided that the input parts are reasonably represented as point clouds, discussed below.
>
> ---
>
> > **Q2**: Following Q1, if one generates some bone fragments, which are likely unseen by the models, how well the models can generate in assembling bones?
>
> We  agree that bone fragments are an important application of 3D shape assembly. To evaluate our method in this context, we conducted zero-shot tests on bone scans from the test split of the newly-released **Fractura** \[11] dataset, which includes human bones (`Leg`, `Hip`, `Vertebra`, `Rib`) and pig bones (`Pig Bones`). Below, we report Part Accuracy (%) for both our model and GARF:
>
>
> | Setting      | Method |    Leg   |    Hip   |    Rib   | Vertebra | Pig Bones |    All   |
> |--------------|-------|:--------:|:--------:|:--------:|:--------:|:---------:|:--------:|
> | *Supervised* | *GARF*  |   *89.7*   |   *80.8*   |   *74.8*   |   *60.8*   |    *79.0*   |   *77.3*   |
> | Zero-shot    | GARF  |   70.5   | **72.8** |   62.9   |   37.7   |    53.4   |   57.7   |
> | Zero-shot    | Ours  | **79.9** |   63.4   | **76.2** | **42.0** |  **63.2** | **64.4** |
>
>
> Our model demonstrates stronger zero-shot performance on previously unseen bone fragments, outperforming GARF in most categories, particularly for Leg, Rib, and Pig Bones. However, there remains a gap compared to fully supervised training. We expect that training on relevant medical datasets would further improve accuracy of our model, as it would allow the model to better learn bone-specific shape priors and fracture geometries.
>
>
> ---
>
> > **Q3**: If one has to train from scratch, is there any insight on the minimal requirements of training set?
>
> This is an excellent question. From our experience, successful training requires a dataset with:
>
> 1. **Shape prior**: A sufficient number of objects covering the final (assembled) shape distribution, and
> 2.  **Overlap geometry**: Informative geometry of the overlapping region, such as fracture region or joints.
>
> For specialized domains, e.g., medical or paleontological reconstructions, collecting a domain-relevant dataset with hundreds to thousands of objects and their fragmentations will be necessary for strong generalization. We will release our model weights, training code, and scripts for fine-tuning on custom datasets to support the community.
>
> ---
>
> > **Q4**: Another challenging scenario (also with strong medical background) is non-rigid shape assembly. Is there any insight on how the proposed framework can be extended to the more challenging setting?
>
> Our current framework is designed for rigid shape assembly, and extending it to handle non-rigid or deformable assembly would require substantial conceptual and methodological changes, such as defining overlap points and learning shape priors over the distribution of all possible non-rigid deformations. We acknowledge this limitation and highlight non-rigid assembly as an important and challenging direction for future research, with broad applications for fields such as robotic manipulation, medical reconstruction, and material reuse.

---

> > ### Comment · Reviewer_yLRg · 2025-08-03
> >
> > Thank you for the response, I remain in the positive side of this paper.

---

### Official Review · Reviewer_q4io · 2025-07-02

**Clarity:** 3
**Significance:** 3
**Originality:** 3
**Rating:** 4
**Confidence:** 3

**Summary:**

This paper proposes Rectified Point Flow (RPF), which is a unified framework for 3D point cloud pose estimation, addressing both pairwise registration and multi-part shape assembly problems as a single conditional generative task. RPF learns a continuous point-wise velocity field to transport noisy points to their target positions, from which poses are then recovered. This approach inherently learns and handles assembly symmetries without explicit labels. Coupled with a self-supervised encoder that focuses on overlapping points, RPF achieves state-of-the-art performance across six benchmarks.

**Questions:**

Experiments:
- The trade-off between time complexity and quality. What the results will be with different iterations of conditional generation.
- The size of the point cloud. Are the point clouds sampled to the same fixed number of point clouds? What is the value of M_i? Will it be different for parts?
- Baseline. Do the baselines also use the pre-trained dataset used in RPF?
- The straight-line interpolation is used for each part (line 153). Does this linear interpolation matches the flow process, which is unlikely to be linear?
- It would be good to see the denoising intermediate results in the main text.
- What is the effect of the overlap self-supervised pre-training? It would be good to see the results with and without the self-supervised task in the main text.

**Ethical Concerns:**

["NO or VERY MINOR ethics concerns only"]

**Final Justification:**

The authors added further analysis and experimental result to address my concerns. They promise to include those in the revised manuscript.

**Limitations:**

Yes. The failure cases were discussed (Fig. 7).

**Quality:**

3

**Strengths And Weaknesses:**

Strengths

- Both pairwise registration and multi-part assembly tasks are tackled in a unified framework of per-part pose estimation with the proposed point flow based point cloud pose estimation method.
- A explicit function of overlap prediction between parts is formulated to facilitate the learning of inter-part relationships.
- The comparison with state-of-the-art shows performance improvement across the evaluated metrics.

Weaknesses

- The experimental comparison against baselines may be biased due to the usage of the pre-trained dataset.
- time consumption. Time complexity comparison is missing.
- others as listed in the questions.

---

> ### Author Rebuttal · Authors · 2025-07-31
>
> We sincerely thank the reviewer for their insightful and constructive feedback and greatly appreciate the recognition of our "unified framework for both pairwise registration and multi-part assembly tasks." Please find our responses to the comments below.
>
> ---
>
> > **Q1** The experimental comparison against baselines may be biased due to the usage of the pre-trained dataset. Do the baselines also use the pre-trained dataset used in RPF?
>
> We would like to point out that in all baseline methods, only GARF proposed pretraining in their method and GARF is the strongest baseline. Other baselines do not require or include pretraining in their methods. For GARF, we use the same pretraining dataset as in RPF for a fair comparison. As shown in Tables 2 and 3, our method outperforms GARF across the board. In our ablation study (Table 4), we also show the effectiveness of our proposed pretraining task.
>
> ---
>
> > **Q2**: The trade-off between time complexity and quality. What the results will be with different iterations of conditional generation.
>
> We thank the reviewer for highlighting this important trade-off. As suggested, the primary variable affecting both runtime and accuracy is the number of sampling steps (iterations). Below, we provide a table summarizing Part Accuracy, Chamfer Distance, and runtime per sample at different sampling steps on PartNet dataset. All tests were performed on a single RTX 4090 GPU.
>
> We find that performance nearly saturates for steps greater than 50, and that the best trade-off balancing speed and accuracy is at 20 steps, which has already been used in all evaluations in the paper.
>
> | Sampling steps               | 1 | 2 | 5 | 10 | 20 | 50 |
> |----------------------|------|-------|-------|--------|--------|--------|
> | Part Accuracy \[%]    |  25.2  | 38.1    | 46.7   | 52.1    |   53.9   |  54.6    |
> | Chamfer Distance \[cm] |  3.23  |  1.70   |  0.90   |   0.75   |  0.73   | 0.71  |
> | Runtime per sample \[s] | 0.072  | 0.081 |  0.108   | 0.148    | 0.232    | 0.483    |
>
> For reference, below is the detailed runtime breakdown for the 20-step setting: most of the runtime is attributed to the flow model, as expected.
>
> | Stage                          | Runtime per sample \[s] | Percentage \[%] |
> |------------------------------- |-----------------------|---------------|
> | Point cloud encoder            | 0.055                 | 23.7          |
> | Flow model with 20 sampling steps    | 0.169                 | 72.8          |
> | Rigid transformation fitting by SVD   | 0.008                 | 3.5           |
> | **Total**                      | **0.232**             | 100.0           |
>
> We will include these results in the revised manuscript.
>
> ---
>
> > **Q3**:  The size of the point cloud. Are the point clouds sampled to the same fixed number of point clouds? What is the value of M_i? Will it be different for parts
>
> In our experiments, we sample 5000 points per object, over all its parts. The number of sampled points $M_i$ for the $i$-th part is proportional to that part’s surface area. This sampling ensures that the flow is equivalent with respect to the assembly symmetry group. Further details can be found in Section E of the Supplementary Material.
>
> ---
>
> > **Q4**: The straight-line interpolation is used for each part (line 153). Does this linear interpolation matches the flow process, which is unlikely to be linear?
>
> We have visualized the flow process of each point at inference time and found that most of the point trajectories are close to straight lines. It is true that, during inference, the generated flows may indeed be nonlinear. This drift (i.e., error accumulation) during inference is a well-known issue for flow-based models \[B, C]. While test-time optimization techniques may help mitigate this, we leave this to future work.
>
> **Reference**
>
> \[B] Liu, Xingchao, Chengyue Gong, and Qiang Liu. "Flow straight and fast: Learning to generate and transfer data with rectified flow." in ICLR 2023.
>
> \[C] Zhou, Zhengyu, and Weiwei Liu. "An Error Analysis of Flow Matching for Deep Generative Modeling." Forty-second International Conference on Machine Learning.
>
> ---
>
> > **Q5**: It would be good to see the denoising intermediate results in the main text.
>
> Thanks for the suggestion. We will include additional flow trajectories and denoising states visualizations to illustrate the intermediate results in the revised manuscript.
>
> ---
>
> > **Q6**: What is the effect of the overlap self-supervised pre-training? It would be good to see the results with and without the self-supervised task in the main text.
>
> We provide this ablation in Table 4 of the main paper, comparing our overlap-based pretraining against no pretraining (random initialization) and semantic segmentation pretraining. Additionally, we also substituted our encoder with PointBERT's pretrained encoder in our response to **Reviewer haCE** (Q2). We found that our overlap-aware pretraining remains the most effective one among all these alternatives evaluated.
>
> Moreover, our pretrained encoder demonstrates particularly strong out-of-domain performance in challenging real-world scenarios such as the 3RScan dataset; see our response to **Reviewer oaRr** (Q1) for details. We will further highlight these results in the revision to clarify the contribution and effectiveness of our pretraining strategy.

---

> > ### Comment · Reviewer_q4io · 2025-08-04
> >
> > Thanks the authors for the detailed response. I will keep my rating positive given most of my concerns are well addressed.

---

### Official Review · Reviewer_haCE · 2025-07-03

**Clarity:** 3
**Significance:** 3
**Originality:** 4
**Rating:** 5
**Confidence:** 4

**Summary:**

This paper presents a unified formulation of point cloud registration and multi-part shape assembly as a conditional generative modeling problem. To address this, the authors propose Rectified Point Flow, a flow-based model that explicitly handles symmetry and part interchangeability—challenges commonly faced in assembly tasks. The method demonstrates strong performance on both shape assembly and registration benchmarks, outperforming prior approaches that rely on deterministic or SE(3)-based optimization.

**Questions:**

- Chamfer Distance in Main Results: Could the authors include the shape Chamfer Distance (CD) in Tables 2 and 4, alongside pose metrics? This would provide a more comprehensive evaluation aligned with the method’s stated goals.
- Comparison with Self-supervised Pretraining: In Table 4, only no pretraining and supervised methods are compared. Please consider including results with standard self-supervised methods (e.g., PointBERT or similar baselines) to contextualize the benefit of the proposed overlap-aware pretraining.
- Overlap Label Symmetry: The overlap supervision seems to assign binary labels to points based on spatial proximity (e.g., only the top region of chair legs is marked as overlapping in Fig. 2). Given that parts like chair legs are symmetric and can be assembled in reverse, does the current labeling overlook valid symmetrical configurations? How does the model handle such ambiguity?
- Anchor Part Selection: Line 118 states that the anchor part is chosen as the first part in the input. Could the authors clarify how the input order is determined? If the order is based on part size or another heuristic, is the assembly result sensitive to different anchor choices?

**Ethical Concerns:**

["NO or VERY MINOR ethics concerns only"]

**Final Justification:**

After reading the rebuttal and the other reviews, my concerns on the evaluation metrics and pre-training are resolved. Therefore, I recommend to accept this paper.

**Limitations:**

Yes

**Quality:**

3

**Strengths And Weaknesses:**

**Strengths**
- Originality: The paper is the first to model point cloud registration and multi-part shape assembly under a shared conditional generation framework. This is a notable departure from prior works that treat the two tasks separately. In contrast to methods like GARF, Rectified Point Flow can naturally handle ambiguous or interchangeable parts.
- Quality & Clarity: The proposed model is thoroughly evaluated through multiple ablations and visualizations. For instance, Figure 4 illustrates how the model scales to larger assemblies, and Table 2 compares both per-task and joint training strategies, clearly showing the benefits of joint learning. Supplementary experiments (e.g., Fig. 11 on cross-category generalization) further strengthen the motivation for using generative modeling for assembly.
- Significance: Addressing part ambiguity and interchangeability is central to robust 3D assembly. This work provides a practical solution through generative modeling in Euclidean space, potentially influencing downstream applications such as robotic assembly or 3D scene synthesis.

**Weaknesses**
- Clarity: While the paper proposes a pretraining approach using overlap prediction, it does not compare this to standard self-supervised pretraining techniques (e.g., PointBERT), leaving unclear whether the proposed strategy is necessary or optimal.
- Clarity: Although the method is motivated by ambiguity and interchangeability, most evaluation metrics in the main paper focus on pose accuracy. Measuring the Chamfer Distance (CD) of the assembled shape would better reflect success under these challenges. While CD is reported in the supplementary (Tab. 5), it is missing from the main tables (Tabs. 2 and 4), making comparisons less complete.

---

> ### Author Rebuttal · Authors · 2025-07-31
>
> We would like to sincerely thank the reviewer for their constructive feedback and suggestions. We are particularly grateful for highlighting our method as "the first to model point cloud registration and multi-part shape assembly under a shared conditional generation framework," which is "a notable departure from prior works." We also appreciate for the comments that our model "explicitly handles symmetry and part interchangeability—challenges commonly faced in assembly tasks," and for noting its potential to "influence downstream applications such as robotic assembly or 3D scene synthesis." Below are our response to the comments.
>
> ---
>
> > **Q1**: Chamfer Distance in Main Results: Could the authors include the shape Chamfer Distance (CD) in Tables 2 and 4, alongside pose metrics? This would provide a more comprehensive evaluation aligned with the method’s stated goals.
>
> Below is the Chamfer Distance (CD, cm, lower is better) metric between our method and GARF over all datasets we evaluated. Our method is consistently better than or on par with the best-performing baseline, GARF across different datasets. We will include the CD numbers in both Tables 2 and 4.
>
> | Task                  | Dataset       | GARF [CD]   | Ours [CD]         |
> |-----------------------|---------------|:-------:|:------------:|
> | Pairwise Registration | TUD-L         |  0.042   | **0.016**     |
> |                       | ModenNet-40   |  0.109   | **0.023**    |
> | Shape Assembly        | BreakingBad-Everyday| **0.025**    | 0.029    |
> |                       | PartNet    |  1.82   | **0.73**     |
> |                       | TwoByTwo   |  2.38   | **0.32**     |
> |                       | IKEA-Manual|  3.28   | **1.96**     |
>
> Please note that the CD values in Table 5 of the Supplementary Material differ due to a different averaging scheme. For a valid comparison with pose-graph methods \[65, 66], we average CD following their convention of assigning each part with the same number of points, whereas in our approach and GARF, the number of points per part is proportional to surface area.
>
> ---
>
> > **Q2**: Comparison with Self-supervised Pretraining: In Table 4, only no pretraining and supervised methods are compared. Please consider including results with standard self-supervised methods (e.g., PointBERT or similar baselines) to contextualize the benefit of the proposed overlap-aware pretraining.
>
> Thanks for this insightful suggestion. We have tested PointBERT [D]’s encoder (dVAE) as a substitute for our own pretrained feature encoder. Following PointBERT’s configuration, we used their encoder weights pre-trained on ShapeNet meshes. Note that ShapeNet and PartNet share the same meshes; PartNet only adds part annotations to a subset of ShapeNet's objects. Using this setup, we trained our flow model on PartNet. The evaluation results are presented in the table below.
>
> | Encoder | Part Acc. \[%\] | Chamfer Dist. \[cm\] | Rot. Err. \[deg\] | Trans. Err. \[cm\] |
> | :---- | :---: | :---: | :---: | :---: |
> | PointBERT Encoder | 45.2 | 1.92 | 27.4 | 23.3 |
> | Our Pretrained Encoder | **53.9** | **0.73** |  **21.8** | **14.8** |
>
> We observe that when using PointBERT’s encoder, the Part Accuracy drops to 45.2%, whereas our pretrained encoder achieves 53.9%, reflecting a 16% relative improvement. Furthermore, localization metrics deteriorate significantly with PointBERT’s encoder — for instance, Chamfer Distance increases from 0.73 cm to 1.92 cm (a 2.6x increase). We attribute these differences to two main reasons:
>
> 1. PointBERT’s encoder is trained for reconstruction on ShapeNet point clouds but is not explicitly optimized for capturing inter-part relationships as our encoder, e.g., overlapping regions.
> 2. The grouping mechanism inside PointBERT’s encoder (with 64 groups by default) limits its ability to preserve fine-grained local geometric details, leading to higher localization errors.
>
> We will include this experiment in Table 4.
>
> **Reference**
>
> [D] Xumin Yu, et al. "Point-bert: Pre-training 3d point cloud transformers with masked point modeling." in CVPR, 2022.
>
> ---
>
> > **Q3**: Overlap Label Symmetry: The overlap supervision seems to assign binary labels to points based on spatial proximity (e.g., only the top region of chair legs is marked as overlapping in Fig. 2). Given that parts like chair legs are symmetric and can be assembled in reverse, does the current labeling overlook valid symmetrical configurations? How does the model handle such ambiguity?
>
> This is a good point. Our overlap labels are based on spatial proximity, which may not capture all symmetrical configurations. However, the flow model itself is designed to account for symmetries based on the per-point feature during training and inference, which can model multiple valid symmetrical configurations beyond what the overlap supervision specifies. More explanation and experiment of the generalization w.r.t. symmetry configurations can be found in our response to **Reviewer JGmh** (Q4).
>
> ---
>
> > **Q4**: Anchor Part Selection: Line 118 states that the anchor part is chosen as the first part in the input. Could the authors clarify how the input order is determined? If the order is based on part size or another heuristic, is the assembly result sensitive to different anchor choices?
>
> During training, part order is randomly shuffled at each iteration for all tasks, so the anchor part is selected randomly. During inference, for the shape assembly task, we select the part with the largest volume as anchor, which is a standard approach for shape assembly, following GARF. For the pairwise registration task, we follow the datasets' pre-defined `target` point cloud as the anchor.

---

> > ### Author Response · Authors · 2025-08-04
> >
> > Dear Reviewer haCE,
> >
> > Thank you again for your constructive  suggestions and feedback—they have been very helpful to us. As the discussion period is still ongoing, we would be glad to discuss any further concerns or questions you may have.
> >
> > Best regards,
> >
> > Authors

---

> > ### Comment · Reviewer_haCE · 2025-08-07
> >
> > Thank you for the detailed rebuttal. It addressed my concerns on evaluation metrics and pre-training schemes. I'll keep my positive rating.

---

### Official Review · Reviewer_oaRr · 2025-07-03

**Clarity:** 3
**Significance:** 2
**Originality:** 3
**Rating:** 4
**Confidence:** 3

**Summary:**

This paper introduces point-cloud registration and part assembly at the object level. The authors propose the Rectified Point Flow to achieve a generalizable point cloud pose estimation scheme and show good results across multiple benchmarks.

**Questions:**

1) How efficient is the overlapping training strategy? All successful results in the paper seem to involve relatively simple parts to be assembled, e.g., the dining table, lamp, etc. However, in the failure case, when the parts become less determinative (e.g., the Wassily Chair), the method fails to assemble them.
2) Is there any performance evaluation on out-of-domain data? I understand that the authors trained across different datasets and used a train/test split, yet there is no demonstration of how distinct the train and test sets are. Especially, there is no out-of-domain evaluation.

**Ethical Concerns:**

["NO or VERY MINOR ethics concerns only"]

**Final Justification:**

Thank you for responding to my questions. The rebuttal addresses most of my concerns, and I am inclined to accept this paper. I recommend that the authors include these discussions and experiments in the final version to further strengthen the work.

**Limitations:**

Yes.

**Paper Formatting Concerns:**

None.

**Quality:**

2

**Strengths And Weaknesses:**

Strengths:
1) The numerical results are good; the proposed method achieves better performance than other counterparts on different datasets.
2) The idea of considering point cloud registration and multi-part shape assembly as a conditional generative problem seems to work on sythentic object-level datasets.

Weaknesses:
1) Method: The overlapping training strategy seems to work only for objects with clear and determinative parts to be assembled, which may be suitable for synthetic toy datasets but not sufficient for most in-the-wild real objects. The claimed 'generic' nature of the method should be toned down.
2) Evaluation: The current evaluation scheme does not fully support the claimed 'generic' performance, as all evaluations are conducted on an in-domain basis. This setup only demonstrates that the Rectified Flow performs better than other baselines in terms of 'remembering' the dataset, rather than validating the proposed method's generalization capability.

---

> ### Author Rebuttal · Authors · 2025-07-31
>
> We sincerely thank the reviewer for the thoughtful feedback, and particularly for highlighting our "novel formulation of pose estimation as a conditional generative process" and recognizing our "strong numerical performance." We appreciate the constructive suggestions, which help us improve the quality of our paper. Below are our responses to the comments.
>
> ---
>
> > **Q1**: The overlapping training strategy seems to work only for objects with clear and determinative parts to be assembled, which may be suitable for synthetic toy datasets but not sufficient for most in-the-wild real objects. The claimed "generic" nature of the method should be toned down.
>
> We appreciate this insightful comment and agree that it is important to address both **(1)** the applicability to real-world meshes and **(2)** the handling of objects with less clearly defined boundaries or overlapping regions.
>
> First, please note that our evaluation in the main paper already includes the TUD-L \[18] dataset, which features real-world 3D scans obtained from depth camera, including partial observations and sensor noise.
>
> To further stress-test our method in this challenging setting of real-world meshes, we conducted additional zero-shot experiments on the **3RScan** \[A] dataset, following the protocol of LivingScenes \[26]. 3RScan consists of room-scale scans captured with real-world RGB-D sensors, and involves relocating objects in a secondary scan, introducing  real-world distributions of noisy points and overlapping regions. Notably, both the dynamic range and noise levels in 3RScan are significantly higher than those in TUD-L. Following \[26], we report the Registraion Recall (%, higher is better) and mean Rotation Error (degree, lower is better). Plus, we report the Translation Error (cm, lower is better).
>
> | Method           | Registration Recall [%] | Rotation Error [deg] | Translation Error [cm] |
> |------------------|:-----------------------:|:--------------------:|:----------------------:|
> | *LivingScenes*    |           *61.1*          |          *7.9*         |            -           |
> | GARF, zero-shot |           0.2          |         38.4         |          26.2          |
> | Ours, with PointBERT encoder, zero-shot |           14.1          |        23.4         |           7.9          |
> | Ours, zero-shot |           **58.9**          |         **11.3**         |           **2.0**          |
>
> From the table, we observe that not only does our method substantially outperform the previous strong baseline, GARF, but it also approaches SOTA performance as established by LivingScenes, without ad-hoc test-time optimization as in LivingScenes.
>
> To further highlight the effectiveness of our overlap-aware encoder and pretraining strategy, we conducted an ablation study where we replaced our encoder with PointBERT's encoder; details in our response to **Reviewer haCE** (Q2). Importantly, in this setting, the flow model is still trained jointly across all datasets, just as in our full model. Nonetheless, performance drops significantly across all metrics, clearly demonstrating that our overlap-aware pretraining design is critical for robust performance in diverse inter-part boundaries and overlap regions.
>
>
> Regarding the term "generic," we acknowledge the need for clarity that we use "generic" to refer to the generic nature of our overlap-aware pretraining across **a wide spectrum of part definitions**. In fact, we train and test our method on six datasets, each with different part definition, specifically:
>
> 1. **TUD-L** \[18\]**:** Part segmentation obtained from real-world 3D scans, which include partial observation and noise.
> 2. **ModelNet** \[19\]**:** Parts segmented by randomly posed planes.
> 3. **BreakingBad** \[12\]**:** Parts are fractured with physical simulation by applying external forces to the object surface;
> 4. **PartNet** \[15\]: Segmentation based on expert-defined hierarchical taxonomies focused on part reusability across objects;
> 5. **IKEA-Manual** \[17\]: Part segmentation defined by functional roles in furniture;
> 6. **TwoByTwo** \[16\]**:** Segmentation annotated for robotic insertion tasks;
>
> This generic encoder features joint training across varied datasets and part definitions, further enhancing performance.
>
> Finally, we will update the manuscript to more clearly scope and define our use of "generic", which is different from "general" and "generalizable".
>
>
> **Reference**
>
> \[A] Johanna Wald, et al. "RIO: 3D object instance re-localization in changing indoor environments." ICCV, 2019.
>
> ---
>
> > **Q2**: How efficient is the overlapping training strategy? All successful results in the paper seem to involve relatively simple parts to be assembled, e.g., the dining table, lamp, etc. However, in the failure case, when the parts become less determinative (e.g., the Wassily Chair), the method fails to assemble them.
>
> As shown in Table 4 of the main paper, we present an ablation study comparing our overlap-aware pretraining strategy against alternatives, including random initialization and pretraining with a semantic segmentation task, where our pretraining yields the best performance. Additionally, as detailed in our response to **Reviewer haCE** (Q2), we substituted our encoder with PointBERT’s encoder (pretrained on ShapeNet) and observed that our overlap-aware pretraining still outperforms this baseline, demonstrating both efficiency and effectiveness in learning part-aware representations.
>
> The assembling of 3D shapes remains a highly challenging problem, particularly for objects with ambiguous or weak geometric cues. On furniture assembly tasks (e.g., PartNet, IKEA-Manual), prior methods such as B-LSTM [66] and RGL-Net [65] struggle with performance and require the object category as part of the input, while our method outperforms them **without** such category input. On fracture reassembly (e.g., BreakingBad), our method is competitive with state-of-the-art methods such as GARF.
>
> For challenging cases like the Wassily Chair, the core difficulty is not just the part definition or overlap points detection, but the inherent lack of discriminative geometric cues. This ambiguity means that the same parts could be plausibly assembled into entirely different objects, making the task under-constrained from geometry alone. We acknowledge this limitation and will include it in the paper that assembly in such cases likely requires additional context information, which is an open problem in the field.
>
> In summary, while all geometric shape assembly methods (including ours) are limited in severely under-constrained cases, our unified point flow formulation, generic pretraining, and symmetry handling advance state-of-the-art performance in real-world scenarios. We appreciate the reviewer highlighting this important direction for future research.
>
> ---
>
> > **Q3**: Is there any performance evaluation on out-of-domain data? I understand that the authors trained across different datasets and used a train/test split, yet there is no demonstration of how distinct the train and test sets are. Especially, there is no out-of-domain evaluation.
>
> We agree that out‑of‑domain (OOD) zero-shot evaluation is valuable. In addition to the zero-shot results on 3RScan [A], in our response to **Reviewer yLRg** (Q2), we also performed zero-shot evaluation on the unseen **Fractura** [11] dataset for the shape assembly task, which includes bone scans from various body parts. Consistent with previous findings, our model outperforms GARF in the zero-shot setting.
>
> We would like to clarify, however, that our paper does **not** claim universal cross-domain generalization or a foundation 3D assembly model. Rather, our focus is on a generic formulation across diverse part definitions, tasks, and symmetries within the domains studied. Generalization to arbitrary OOD data or developing a foundation assembly model remains an open challenge for future work.

---

> > ### Author Response · Authors · 2025-08-04
> >
> > Dear Reviewer oaRr,
> >
> > Thank you again for your constructive comments—they have been very helpful to us. As the discussion period is still ongoing, we would be glad to discuss any further concerns or questions you may have.
> >
> > Best regards,
> >
> > Authors

---

> > > ### Comment · Reviewer_oaRr · 2025-08-06
> > > **Official Comment by Reviewer oaRr**
> > >
> > > Dear authors, thank you for addressing my questions. Your rebuttal resolves most of my concerns, and I will be increasing my score accordingly.

---

### Note · Authors · 2025-08-13

We sincerely thank all reviewers for their constructive feedback and discussion. We are encouraged by the broad agreement that our method **(i)** offers a novel formulation unifying pairwise registration and shape assembly as a single conditional generative task, **(ii)** intrinsically handles symmetry and part interchangeability, and **(iii)** achieves SOTA across diverse benchmarks.

---

During the rebuttal, we addressed the concerns from all the reviewers, particularly regarding:

**Scope**: By “generic,” we mean the framework can be jointly trained across datasets with diverse part definitions (e.g., random, functional, fractured, scanned, and semantic). This enables shared geometric prior learning across different object structures and leads to robust performance.

**Generalization**: To address concerns about out-of-domain performance, we conducted additional evaluations:

  - **3RScan** (pairwise registration in real-world room scans): Our method achieves 58.9% registration recall in zero-shot despite significant sensor noise and partial observations, only second to the SOTA approach LivingScenes' 61.6%.

  - **Fractura-Bone** (shape assembly on an unseen medical domain): Our method achieves 64.4% part accuracy in zero-shot, outperforming GARF's 57.7% and demonstrating real-world applicability beyond man-made objects.

  - **Cylindrical toy dataset** (shape assembly with novel symmetry types): Our method demonstrates high part accuracy (97%+) even when trained only on horizontal cuts and tested on novel axial cuts and random 3D cuts, confirming strong generalization on unseen symmetry types.

**Pretraining**: Beyond the ablations in the main paper, we compared against PointBERT’s encoder. Our overlap-aware pretraining achieves higher part accuracy (53.9% vs. 45.2%) and lower Chamfer distance (0.73 cm vs. 1.92 cm) on PartNet, along with stronger cross-domain generalization to 3RScan. This indicates our pretraining better learns inter-part relationships than standard reconstruction-based approaches.

---

Following the rebuttal, all five reviewers responded positively, with at least two increasing their ratings further.

In summary, we hope that our unified **_pose-from-shape_** formulation with the **_symmetry-aware generative model_** advances point cloud understanding and inspires new research in robotic assembly, 3D reconstruction, and medical applications. We will include all suggested edits and additional evaluations in the revised paper.

---

### Decision · Program_Chairs · 2025-09-17

**Decision:**

Accept (spotlight)

**Comment:**

This paper received positive ratings from all reviewers; all reviewers agree that the paper tackles an important problem, proposes an interesting approach, and shows significant improvement over previous SOTA. The rebuttal also addresses most of the reviewer’s concerns, in particular, the generalization to unseen datasets. The AC also appreciates the scientific contribution of unifying registration and assembly in a novel conditional generative task, and thus recommends accepting this work.